# Shifts in stream hydrochemistry in responses to typhoon and non-typhoon precipitation

Chung-Te Chang[1], Jr-Chuan Huang[1], Lixin Wang[2], Yu-Ting Shih[1], and Teng-Chiu Lin[3]

[1]Department of Geography, National Taiwan University, No 1 Section 4, Roosevelt Road, Taipei 10617, Taiwan
[2]Department of Earth Sciences, Indiana University-Purdue University Indianapolis, Indianapolis, IN 46202, USA
[3]Department of Life Science, National Taiwan Normal University, No 88 Section 4, Ting-Chow Road, Taipei 11677, Taiwan

*Correspondence to*: Teng-Chiu Lin (tclin@ntnu.edu.tw)

**Abstract.** Climate change is projected to increase the intensity and frequency of extreme climatic events such as tropical cyclones. However, few studies have examined the responses of hydrochemical processes to climate extremes. To fill this knowledge gap, we compared the relationship between stream discharge and ion input-output budget during typhoon and non-typhoon periods in four subtropical mountain watersheds with different levels of agricultural land cover in northern Taiwan. The results indicated that the high predictability of ion input-output budgets using stream discharge during non-typhoon period largely disappeared during the typhoon periods. For ions such as $Na^+$, $NH_4^+$, and $PO_4^{3-}$, the typhoon period and non-typhoon period exhibited opposite discharge-budget relationships. In other cases, the discharge-budget relationship was driven by the typhoon period, which consisted of only 7% of the total time period. The striking differences in the discharge-ion budget relationship between the two periods likely resulted from differences in the relative contributions of surface runoff, subsurface runoff and groundwater, which had different chemical compositions, to stream discharge between the two periods. Watersheds with a 17–22% tea plantation cover showed large increases in $NO_3^-$ export with increases in stream discharge. In contrast, watersheds with 93–99% forest cover showed very mild or no increases in $NO_3^-$ export with increases in discharge and very low levels of $NO_3^-$ export even during typhoon storms. The results suggest that even mild disruption of the natural vegetation could largely alter hydrochemical processes. Our study clearly illustrates significant shifts in hydrochemical responses between regular and typhoon precipitation. We propose that hydrological models should separate hydrochemical processes into regular and extreme conditions to better capture the whole spectrum of hydrochemical responses to a variety of climate conditions.

## 1 Introduction

One of the major concerns of global climate change is increases in extreme climatic events such as flooding, droughts, and tropical cyclones (Phillips, 2017). Mounting evidence suggests that such events have strong effects on ecosystem function such as biodiversity, productivity, phenology, nutrient cycling, and community resistance to invasion (Holmgren et al., 2006; Fay et al., 2008; Jentsch and Beierkuhnlein, 2008; Smith, 2011; Chang et al., 2017a; Sinha et al., 2017). Predicting ecological effects of climate extremes is challenging because their effects on ecosystems could be dramatically different from "typical" or "normal" climatic variability (Smith, 2011).

Land use change has been considered a potential environmental threat at both local and global scales (Foley et al., 2005; Tang et al., 2005). A large number of studies have reported that replacing natural forests with agriculture lands causes large increases in surface runoff, sediment yield and nutrient export (Kosmas et al., 1997; Hill et al., 1998; Gessesse et al., 2015). Locally, in a study of nutrient cycling in upstream watersheds of northern Taiwan, the replacement of 22% of the natural forests by tea plantations reduced the nitrogen retention ratio by 50% (Lin et al., 2015). The consequences of land use change on nutrient retention is likely most dramatic during extreme events such as tropical cyclones when precipitation exceeds soil infiltration capacity. A study on paired watersheds in Taiwan indicated that sediment yield was one order of magnitude lower in plantations with gentler slopes than natural forests with steeper slopes during base flow (Tsai et al., 2009). However, during the peak flow of a typhoon event, the sediment yield was one order of magnitude greater in the plantations than the natural forests (Tsai et al., 2009).

Studies of nutrient input and output in both temperate and subtropical regions reported that hydrological control of the net nutrient input-output budget could override the effect of plant growth, leading to greater nutrient export in the growing season when biological demand is high (Likens and Bormann, 1995; Chang et al., 2017a). Although rarely examined, it can be expected that differences in nutrient export between disturbed and undisturbed watersheds are most dramatic during extreme storm events, relative to less extreme, typical periods.

With the projected increases in climate extremes in many parts of the world (Elsner et al., 2010; Donat et al., 2016; Borodina et al., 2017; Pfahl et al., 2017), the relationship between precipitation or stream discharge and nutrient export could shift to a new phase, which cannot be extrapolated from relationships that are mostly driven by "typical" storms. In a previous study, we illustrated differences in monthly nutrient input and output among four mountain watersheds differing in levels of tea plantation cover in northern Taiwan (Lin et al., 2015). Here, we report the differences in the ion input-output budget between "regular" flow periods and typhoon periods in the four watersheds. The objectives of this study are to 1) test if typhoon storms will cause distinct alternation in nutrient input-output budget due to the nonlinear nature of many ecological processes in response to disturbance (Burkett et al., 2005; Jentsch, 2007); and, 2) to examine differences in the relationship between stream discharge and input-output budget among ions and among watersheds with different levels of agricultural land cover.

## 2  Materials and Methods

### 2.1  Study region

This study was conducted at the 303 km$^2$ subtropical Feitsui Reservoir Watershed (FRW) in northern Taiwan (Fig. 1a). The FRW region is characterized with humid subtropical climate. The mean annual precipitation is 3765 mm between 1991 and 2001 (Chen et al., 2006), with approximately 68% occurring between May and September (Chang and Wen, 1997). However, due to the rough topography, precipitation is highly variable, ranging from 3500 mm in the southwest portion of the FRW to more than 5000 mm in the northeast during 2001–2010 (Huang, C. J., unpublished data). This area is covered mostly by natural secondary forests dominated by tree species within the Fagaceae and Lauraceae families (Chen, 1993). Because the FRW is a

water resource protection area, agricultural activities are limited to pre-existing agriculture lands, mostly tea plantations (1200 ha). Tea plantations comprise approximately 15.8% of the FRW (Chang and Wen, 1997; Chou et al., 2007). Fertilizer applications are heavy in the tea plantations, reaching 786 kg-N ha$^{-1}$ yr$^{-1}$ (Lin et al., 2015). The FRW has a rough topography with an elevation ranging from 45 m to 1127 m and a mean slope of 42% (Fig. 1a and 1b). Soils in the FRW are mostly Entisols

and Inceptisols with high silt contents developed from argillite and slate with sandstone interbeds (Zehetner et al., 2008).

## 2.2  Sampling scheme

We sampled stream water at four subwatersheds (A1, A2, F1, and F2) and precipitation water at two of the four subwatersheds (A1 and F2) within FRW on a weekly basis between September 2012 and August 2015 (Fig. 1a). Natural forest is the major land cover type of all watersheds (> 68%); however, agricultural lands are also important at A1 (22%) and A2 (17%). A1, A2,

and F2 are small watersheds (< 3 ha) drained by first order streams while the F1 watershed (86-ha) is drained by a third-order stream that drains through A1 and A2 (Fig. 1).

Weekly samples were collected with a 20-cm diameter polyethylene (PE) bucket. Weekly stream water samples were collected by immersing a PE bucket into the stream. For both precipitation and stream water, a 600 mL subsample was taken using a PE bottle and transported to the laboratory with conductivity and pH being measured the same day of collection. After

the measurement of pH and conductivity, samples were filtered (0.45 μm filter paper) mostly within eight hours of sample collection. All the filtered samples were stored at 4$^{o}$C without chemical preservatives prior to chemical analysis. Concentrations of major cations (Na$^+$, K$^+$, Ca$^{2+}$, Mg$^{2+}$, NH$_4^+$) and anions (Cl$^-$, SO$_4^{2-}$, NO$_3^-$) were analyzed by ion chromatography on filtered samples using Dionex ICS 1000 and DX 120 (Thermo Fisher Scientific Inc. Sunnyvale, CA, USA). PO$_4^{3-}$ concentration was measured using the standard vitamin C-molybdenum blue method with a detection limit of 0.01 μM

(Rice et al., 2012).

## 2.3  Precipitation and stream flow estimation

Precipitation in mountainous area is quite dynamic due to the interaction between orography and circulation. Following Huang et al. (2011), we used 10 rainfall stations to simulate the discharges of the four sites via Hydrologiska Byråns Vattenbalansavdelning (HBV) model. The areal rainfall from Thiessen polygon was applied, and thus the rainfall spatial

heterogeneity has been considered partially. Precipitation of each of the four watersheds was then obtained from the spatial distribution of precipitation. Stream discharge of the four ungauged watersheds was also simulated by the HBV model processed through TUWmodel (ver. 0.1-8) (Parajka et al., 2013). Five daily rain gauges, maintained by Water Resource Agency (WRA), and five metrological stations, maintained by the Central Weather Bureau (CWB) of Taiwan with hourly observed rainfall, temperature, wind speed, and solar radiation were used to estimate daily rainfall and potential

evapotranspiration. The daily evapotranspiration is also observed by Taipei Feitsui Reservoir Administration (TFRA, Taiwan) at the Feitsui meteorological station. The observed rainfall, temperature and evapotranspiration were applied into 20 sub-catchments with Thiessen polygon method. Daily discharge was monitored in three main tributaries of Baishi Creek by TFRA.

In the calibration against the observed values, parameters were generated by the package DEoptim (ver. 2.2-4) (Mullen et al., 2011). Three objective functions, Nash Efficient Coefficient (NSE), its power of 2 and log scale, were used to adjust the model to suit normal, extreme, and low flow conditions. The total runoff derived from the HBV model was further separated into three components, surface runoff, subsurface runoff and groundwater. The validation gauge is located in the inflow of dam of reservoir. The modelled daily discharge was aggregated into weekly discharge. Although the HBV model has been successfully applied in northern Taiwan (Chang et al., 2017a), due to the lack of in-situ measurements of discharge, the estimates are subject to some uncertainty. The paired weekly ion concentrations and water volume of precipitation and streamflow were used for the ion input-output budget calculations (i.e., output via stream discharge – input through precipitation).

## 2.4 Definition of typhoon-affected samples and large non-typhoon samples

Because we did not sample precipitation and stream water on a storm-by-storm basis, we separated the weekly samples into typhoon samples and non-typhoon samples to examine the effects of typhoon storms on hydrochemistry. Following Chang et al. (2013), weekly samples collected between the first and last typhoon warnings issued by the CWB of Taiwan are considered typhoon samples, and such a week was referred as a typhoon-affected week. Although there is a time lag between precipitation and streamflow, this lag was typically only a few hours in mountain watersheds of Taiwan (Huang et al., 2012), so this short lag has only limited effects on the division of typhoon and non-typhoon samples. This definition may overestimate the total quantity of precipitation and stream discharge associated with typhoon storms because typhoons rarely lasted for a week; thus, part of the weekly samples classified as typhoon samples included water before or after the typhoon storm periods. In contrast, this definition diluted the extreme nature of typhoon storms, as the weekly samples included some water from small storms or base flow. Although a storm-based sampling would better capture the effects of typhoon storms on hydrochemistry, it is dangerous to collect samples during typhoons, and it would also miss the base flow hydrochemistry. To compare discharge-ion budget relationship between typhoon periods and periods with precipitation comparable to typhoon weeks, we identified seven weeks that had precipitation greater than precipitation of the minimal typhoon storms, 160 mm and categorized them as large non-typhoon precipitation weeks.

## 3 Results

### 3.1 Basic storm information

During the sampling period, weekly precipitation ranged from 1 mm to 507 mm while weekly streamflow ranged from 10 mm to 446 mm (Fig. 2a and Table S1). The weekly runoff ratio was negatively related to precipitation quantity and was highly variable during the non-typhoon period but varied much less during the typhoon period (Fig. 2b). The ratio of total runoff to precipitation was not different between non-typhoon period (0.69–0.81) and the typhoon period (0.64–0.78) but the ratio of surface runoff to precipitation was smaller in the non-typhoon period (0.06–0.15) than the typhoon period (0.27–0.33) (Table 1) because proportionally surface runoff was greater during the typhoon period than the non-typhoon period (Fig. 3). There

was a total of 11 typhoon-affected weeks based on our definition. The 11 typhoon-affected weeks contributed 2862 mm or 26% of total precipitation (10845 mm) and 1991 mm or 22% of total stream discharge (9067 mm) for the three sampling years (Fig. 2 and Table S1). The quantity of precipitation and discharge of typhoon-affected weeks ranged from 168 and 122 mm for typhoon Goni (21-24 August 2015) to 507 and 446 mm for typhoon Soudelor (7-9 August 2015), respectively (Table S1).

Typhoons contributed 87–98% of the weekly precipitation and 80–93% of the weekly discharge, respectively, of the 11 typhoon-affected weeks (Table S1). The mean weekly precipitation ($\pm$ standard deviation) for the typhoon-affected weeks, 278 ($\pm$ 96) mm, was approximately 4.6 times of that for the non-typhoon weeks, 61 ($\pm$ 64) mm. The mean weekly stream discharge for the typhoon-affected weeks, 210 ($\pm$ 88) mm, was approximately 3.7 times of that for the non-typhoon weeks, 57 ($\pm$ 49) mm.

The weekly maximal hourly, 6-hr, 12-hr and 24-hr precipitation of the typhoon-affected weeks were generally considerably greater than those of the non-typhoon weeks and the differences were greater with greater time intervals. The greatest value of maximal hourly, 6-hr, 12-hr and 24-hr precipitation during the typhoon period reached 54, 43, 33, and 19 mm hr$^{-1}$, respectively, based on the records in rain gauge COA530 (Fig. 4). Among the 10 highest hourly, 6-hr, 12-hr and 24-hr precipitation events, 5, 8, 9, and 9 of them occurred during weeks associated with typhoon storms (Fig. 4).

## 3.2   Stream discharge as a predictor of watershed ion export

One striking pattern during the typhoon period (i.e., the 11 typhoon-affected weeks) is the lack of predictability of stream discharge on input-output budget for many ions in most watersheds. This lack of predictability is in contrast to the high level of predictability during the non-typhoon period (Figs. 5, 6 and Table S2). During non-typhoon period, stream discharge is a good predictor of net export of all ions except $NH_4^+$ for all watersheds, and for $NO_3^-$ in F2 and $PO_4^{3-}$ in A1, A2 and F2 (Figs.

5 and 6). In contrast, during the typhoon period, discharge was not a significant predictor for 14 of the 36 ion budgets (4 watersheds x 9 ions) (Figs. 5, 6 and Table S2). In addition to the low predictability, variability in input-output as indicated by their standard errors was several times greater during large non-typhoon precipitation weeks and typhoon weeks, relative to non-typhoon period (Fig. 7).

## 3.3   Differences between typhoon and non-typhoon periods

In addition to the lack of predictability of stream discharge for input-output budgets during typhoon periods, there were distinct differences in the discharge-budget relationship between typhoon and non-typhoon periods for many ions. There was a positive relationship between stream discharge and the $Na^+$ budget during the non-typhoon period for all watersheds, with greater discharge associated with greater net $Na^+$ export in all watersheds (Fig. 5). However, the relationship was negative during the typhoon period for all watersheds except A1, with greater discharge associated with greater $Na^+$ retention (Fig. 5). There was

also a positive relationship between stream discharge and $PO_4^{3-}$ budget during the non-typhoon period for watershed F1, but during the typhoon period the relationships were negative except for F2 (Fig. 6). The distinct difference between the two

periods was also reflected in the overall net export of $K^+$ during the non-typhoon periods and net retention during the typhoon periods for F2 (Fig. 5).

In addition to the opposite directions of the relationship between discharge and ion budget between typhoon and non-typhoon periods, the 11 typhoon-affected weeks also affected the overall relationship between discharge and ion budget. The positive relationship between discharge and $Cl^-$ budget during the non-typhoon period disappeared in all watersheds when the 11 typhoon-affected weeks were included in the analysis (Fig. 6). Similarly, the positive relationship between discharge and $Na^+$ budget in watersheds A1 disappeared when the typhoon-affected weeks were included (Fig. 5). In contrast, the relationship between discharge and $NH_4^+$ budget changed from non-significant during the non-typhoon period to significantly negative during the typhoon period (Fig. 5). For the $PO_4^{3-}$ budget of F1, including the typhoon-affected weeks changed the relationship from positive to negative (Fig. 6). The budget of most ions of the seven large non-typhoon storms, with precipitation greater than the minimum typhoon precipitation (160 mm) was between the budget of typhoon weeks and regular non-typhoon weeks, but there were fundamental differences (Fig. 7). For example, the negative budget of $Na^+$, $Cl^-$ and $PO_4^{3-}$ was only observed during typhoon weeks (Fig. 7).

### 3.4 Differences among watersheds with different proportions of agricultural land

Nitrate exhibited a unique pattern in the relationship between stream discharge and input-output budget. Stream discharge was an excellent predictor of net $NO_3^-$ export in watersheds A1 and A2 during non-typhoon period with $R^2$ of linear regression of 0.93 in A1 and 0.91 in A2 (Fig. 6 and Table S2). Although there was also a significant positive relationship between stream discharge and $NO_3^-$ budget (net export) during non-typhoon period in F1, the predictability was considerably lower ($R^2 = 0.23$) than those of A1 and A2 (Fig. 6). For watershed F2, stream discharge was not a significant predictor of $NO_3^-$ input-output budget during non-typhoon periods (Fig. 6). The mean input-output budget of F1 and F2 was an order of magnitude lower than that of A1 and A2, and the mean weekly $NO_3^-$ budget of F2 was only -0.01 kg ha$^{-1}$ w$^{-1}$, which was not different from zero (p = 0.881).

The input-output budget of $PO_4^{3-}$ and $K^+$ was also dramatically different among the four watersheds (Figs. 5 and 6). During non-typhoon periods there was a positive relationship between $K^+$ budget and stream discharge with greater net export associated with greater discharge in all watersheds. However, this relationship was not significant in A1, A2 and F1 during typhoon periods and there was a significantly negative relationship during typhoon periods at watershed F2 (Fig. 5). Moreover, the typhoon-affected weeks changed the relationship from positive (without typhoon data) to negative (with typhoon data) for watershed F2 (Figs. 5 and 6). In addition, the mean weekly budget of watersheds A1 and A2 was greater during typhoon periods than during non-typhoon periods, but for watersheds F1 and F2 this budget was greater during non-typhoon periods than during typhoon periods (Figs. 5 and 6). There was a negative relationship between stream discharge and $PO_4^{3-}$ budget during the typhoon-affected period at watersheds A1, A2 and F1, with greater discharge associated with greater retention, but this relationship disappeared at F2, which had a weekly budget of $PO_4^{3-}$ near zero at F2 (Fig. 6). In addition, there was an overall net retention during typhoon periods at all watersheds except F2 (Fig. 6).

For K[+], there were positive relationships between discharge and weekly input-output budgets across different watersheds for non-typhoon periods; however, the $R^2$ and the slopes of the regression lines decreased with increases in forest cover, from 0.65 and 0.009 in A1 to 0.30 and 0.003 in F2 (Fig. 5 and Table S2). An opposite pattern was found for Na[+] during the non-typhoon periods, with the $R^2$ increased with increases in forest cover (Fig. 5 and Table S2).

## 4 Discussion

### 4.1 Differences between typhoon and non-typhoon periods

The striking differences in the discharge-budget patterns between typhoon and non-typhoon periods should be related to changes in the relative proportion of sources of stream discharge. Stream discharge originates from three sources, surface runoff, subsurface runoff and groundwater. Among the three sources, groundwater was more important during low than high

flow periods, whereas the contribution from surface runoff should be more important during heavy storms than small storms. The contribution from subsurface flow probably dominated the discharge at our study site, especially in F1 and F2 because a study at a natural forest 12 km Southeast from our study site indicated that even during a heavy typhoon storm, with precipitation near 700 mm in two days, there was no observable surface runoff (Lin et al., 2011). The contribution from subsurface runoff and groundwater to total discharge likely resulted in the very high runoff ratios for weeks with small amount

of precipitation. For example, in 28 January 2014, the weekly precipitation and discharge were 1.5 mm and 13 mm, respectively, which led to the highest runoff ratio, 8.7, for the entire study period (Fig. 2). The effect of subsurface runoff and groundwater on disrupting the precipitation-runoff relationship is evident from the greater ratio of surface runoff to precipitation during the typhoon period than the non-typhoon period while the ratio of total runoff to precipitation was not different between the two periods (Table 1 and Fig. 3). Our results indicate that under certain circumstances, contributions from baseflow need to be

removed in order to detect and meaningfully assess the precipitation-runoff relationship (Table 1). However, it is noted that without direct measurements of streamflow in two of the watersheds, it is difficult to confidently validate the estimation of streamflow separation in this case.

Changes in relatively contributions of different sources of water (or old water relative to new water) on stream discharge, play a key role in regulating ion concentrations during a storm and between periods of different flow rates (Elwood and Turner,

1989; Giusti and Neal, 1993; Bishop et al., 2004). Among the three sources, groundwater is enriched with ions derived from rock weathering such as K[+], $Ca^{2+}$, and $Mg^{2+}$, pre-storm subsurface runoff have a longer contact time with soils that are also rich in these cations and $SO_4^{2-}$, and surface runoff largely reflect precipitation chemistry. A study of storm solute transport in a forested watershed in northern Taiwan, 12 km Southwest of our study site, indicated that concentrations of Na[+], $Ca^{2+}$, $Mg^{2+}$, Cl[-], and $SO_4^{2-}$ were diluted due to the mixing of large quantities whereas concentrations of K[+], $NH_4^+$ and $NO_3^-$ were enhanced

during high flows (Wang et al., 1996). In our study, the greater contributions from groundwater and subsurface runoff in the non-typhoon period likely contributed to the greater (more positive) slopes between discharge and budget of many ions for the non-typhoon period than typhoon period, in which many of the relationships were not significant (Figs. 5 and 6). The second

possible reason for the greater slopes between discharge and budget of many ions during the non-typhoon period is the differences in ion concentration between typhoon and non-typhoon storms. The day or two days before a typhoon typically has clear sky because the outskirt air masses of the typhoon "blow" away most air pollutants. As a result, precipitation associated with typhoons have low concentrations of ions with terrestrial sources (Lin et al., 2011). In our study, mean concentrations of all ions were lower during typhoon period than non-typhoon period (Table S3) and this diluted precipitation ion concentrations overrode quantity effect and contributed to the smaller increases in budget with increasing discharge in the typhoon period than the non-typhoon period (Figs. 5 and 6).

## 4.2 Unpredictability of hydrochemical response to climate extremes

The large differences in weekly precipitation, stream discharge and weekly maximal hourly, 6-hr, 12-hr and 24-hr precipitation between the typhoon and non-typhoon periods (Table S1) clearly illustrate the extreme effects of typhoon storms. The lack of predictability of stream discharge on ion input-output budgets for the typhoon period is attributable to the high variability of ion budgets associated with typhoon storms (Figs. 5, 6 and Table S2). High variability associated with typhoons is not limited to ion budgets but also to water resources. In a study of long-term biogeochemistry in a natural hardwood forest in northeastern Taiwan, the 20-year average annual precipitation was 3840 mm, but was 3240 mm when precipitation associated with typhoon storms was excluded, with annual contributions from typhoon storms varying from 0% (0/2770 mm) in 1995 to 42% (1711/4033 mm) in 2008 (Chang et al., 2017b).

The lack of predictability of stream discharge on the budget of several ions is possibly due to damages to the forests and farms by the typhoons. Damages to trees may affect the level of foliar nutrient leaching and nutrient uptake by roots and thus the nutrient export (Lin et al., 2011). The poor correlation between maximum wind velocity and precipitation quantity reported by Lin et al. (2011) suggests that precipitation quantity is not a good predictor of the magnitude of typhoon influences on nutrient input-output budget and likely contributed to the low predictability of discharge on ion budget during typhoon period.

Many hydrological models are constructed primarily based on non-extreme conditions or on a combination of both extreme and non-extreme conditions (Wade et al., 2006; Shih et al., 2016; Lu et al., 2017). However, our results showed that in many cases such models would not perform well during extreme conditions such as during typhoons. There are at least three ways that extreme events could lead to model failure. First, in many cases the pattern seen during the more regular period does not exist during extreme conditions, such as a loss of predictability of the budgets of many ions when using stream discharge during the typhoon period (Figs. 5 and 6). Second, in some cases such as the budget of $Na^+$ the models built using non-typhoon data would mistakenly predict the patterns to be in the wrong direction in extreme conditions (Fig. 5). Third, in other cases, the pattern revealed by the models may be driven by only several extreme events, as evident from the cases of $NH_4^+$ at F2, and $PO_4^{3-}$ at A1, A2, and F1 (Figs. 5 and 6).

Climate change will increase the frequency and intensity of extreme climate events such as flooding, drought, and tropical cyclones (Emanuel, 2005; Elsner et al., 2010; Hirabayashi et al., 2013; Cook et al., 2015; Pfahl et al., 2017). Many studies report increases in extreme precipitation events and flooding from observations over the past half century, particularly in the

tropics and subtropics (Hirabayashi et al., 2013; Fischer and Knutti, 2016). Furthermore, the upward trends in frequency and intensity of tropical cyclones due to warming climate is expected to lead to the development of more destructive cyclones (Emanuel, 2005; Elsner et al., 2010). A recent analysis indicated that there was a manifest westward shift of tropical cyclones in the Northwest Pacific (Wu et al., 2015), such that risks of extreme precipitation and flooding events are expected to rise in

this region, which includes Taiwan. Our results show that hydrological consequences of extreme events can not be directly extrapolated from non-extreme conditions. Because rare but extreme events can cause abrupt changes (Müller et al., 2014), separation of hydrochemical processes into more regular and extreme conditions is more likely to capture the whole spectrum of hydrochemical responses to a variety of climate conditions. In addition, regime shifts could invalidate future predictions calibrated on past trends (Müller et al., 2014). Thus, hydrological models must recognize and incorporate the unpredictability

and even chaotic nature of extreme storms to make model predictions more reliable.

## 4.3   Extreme storms intensify the impact of land use change

The differences in the input-output budgets of $NO_3^-$, $PO_4^{3-}$, and $K^+$, the three elements that are the primary ingredients of commercial fertilizers, between A1, A2 and F1, F1 illustrate the effects of replacing natural vegetation with agricultural land use on watershed hydrochemistry. The differences for $NO_3^-$ are most striking. Throughout the three-year period, the budget is

close to zero for F1 and F2 (Fig. 6) illustrating the very high hemostasis of forested watersheds. In contrast, although tea plantations cover only 22% and 17% of the area of A1 and A2, respectively, the dramatic increases in $NO_3^-$ export illustrate that the effects of replacing natural vegetation by agricultural land use is intensified during heavy storms. The results also illustrated that hydrochemistry is distinctly different between forested and agricultural watersheds. More importantly, our results indicate that even mild changes (22% or less) of the land use could have profound effects on critical ecological processes.

Many studies have illustrated negative ecological consequences caused by replacing natural vegetation with agricultural land use (Howarth et al., 2012; Michalak et al., 2013), but few studies have examined the impact during extreme storms when the impact could be maximized. Our results illustrate that when natural vegetation is intact as in the case of F2, hydrochemical processes are relatively stable even during most intense storms (Figs. 5 and 6).

    The close relationship between stream discharge and $NO_3^-$ export from watersheds A1 and A2 across a wide range of stream

discharge (Fig. 6) highlights hydrological control on nutrient export. This implies that there was an ample $NO_3^-$ supply from sources other than precipitation input, which can mostly, if not entirely, be attributed to heavy fertilization, amounting > 700 kg-N ha$^{-1}$ yr$^{-1}$ at our study sites (Lin et al., 2015). This level of fertilization deposits large quantities of $NO_3^-$ in the watersheds, greater than what could be removed by all except the most extreme storms associated with Typhoon Sudelor (2015) leading to the proportional increases in $NO_3^-$ export with increases in stream discharge seen here (Fig. 6). It is important to note that the

weekly export of $NO_3^-$ from watershed A1 reached 40 kg ha$^{-1}$ w$^{-1}$. The consequences of such a high nitrogen input rate to these aquatic systems deserve further investigations.

### 4.4 Unexpected ion retention

The increases in net retention of $NH_4^+$ (all watersheds), $PO_4^{3-}$ (A1, A2, and F1), $Na^+$ (A2, F1, and F2), and $K^+$ (F2) with increases in stream discharge during the typhoon period are unexpected. The pattern of $PO_4^{3-}$ at A1, A2 and F1 watersheds, $K^+$ at F2 watershed, and $Na^+$ at all watersheds were largely driven by a single extreme event which had an extremely negative budget (i.e., output much smaller than input) (Figs. 5 and 6). This extreme event was due to Typhoon Sudelor (2015), which set a new record for wind speed (237 km $hr^{-1}$) in northern Taiwan, causing a serious deterioration in household water quality in Taipei (Taiwan's largest city) (Fakour et al., 2016). This storm also caused a record of sidewalk tree mortality in northern Taiwan. It is not clear if this extreme storm affected the measurement of precipitation and stream flow or damaged the forest to the level that changed the hydrochemical processes in previously unknown ways. However, we could not find any good reason to exclude the data because neither the concentration nor the precipitation: discharge ratio of the week was an outlier. It is the product of the two that makes it significantly different from the general pattern of other typhoon-affected weeks. Thus, perhaps the input-output budget of this most extreme week represents a hydrochemical process phase-shift between the regular extreme events and the most extreme event, as such, the response of the most extreme event cannot be predicted using the data from the regular extreme events. In a study of the effects of typhoons on forest leaf area index in northeastern Taiwan, leaf area index could return to pre-typhoon levels within one year but the decreases associated with a record high number of six typhoons within a year (1994) took approximately one decade to recover (Lin et al., 2017). In other words, the ecological and hydrological effects of record-setting extreme events could be fundamentally different from "regular extreme" events. The most extreme climate events often attract attention and much research has been conducted on them, but based on the current study, results from these studies should be interpreted with caution as they may not represent the overall patterns of extreme events.

However, the greater retention of $NH_4^+$ at all watersheds cannot be attributed to the single week associated with Typhoon Sudelor because the pattern persisted even after this event was excluded (p = 0.042). The pattern of $NH_4^+$ retention was caused by the smaller slope of output vs. discharge compared to input vs. discharge, which was possibly related to slow nitrification rates during extremely large storms.

### 5 Conclusions

Our analysis of ion input-output budget illustrates that hydrochemistry during typhoon storms are highly variable, and models built from regular periods have low predictability of ion budgets during extreme storm periods. Hydrochemical responses to typhoon storms are distinctly different from those of regular storms and have the potential to dominate the long-term hydrochemical patterns. Much greater increases in $NO_3^-$ export associated with increases of stream discharge at watersheds with 17–22% agricultural land cover, relative to $NO_3^-$ exports in watersheds with 93–99% forest cover, indicate that even mild land use change may have large impacts on hydrochemical processes. Climate change is predicted to increase the intensity and

frequency of climate extremes. Based on the results of this study, we suggest separating hydrochemical processes into regular and extreme conditions to better capture the whole spectrum of hydrochemical responses to a variety of climate conditions.

## 6 Data availability

All observational data used in this study necessary to compare with or to reproduce the work are available on request from the corresponding author Teng-Chiu Lin (tclin@ntnu.edu.tw).

*Author contributions.* T.C. Lin designed and performed the research. C.T. Chang, J.C. Huang, Y.T. Shih, and T.C. Lin conducted the field and laboratory work. C.T. Chang, Y.T. Shih and T.C. Lin analysed the data. C.T. Chang, J.C. Huang, L. Wang and T.C. Lin contributed to the discussion and interpretation of the results. C.T. Chang and T.C. Lin wrote the first draft and all authors contributed substantial edits.

*Competing interests.* The authors declare that they have no conflict of interest.

*Acknowledgements.* This study was supported by grants from Ministry of Science and Technology (MOST 101-2116-M-003-003, 102-2116-M-003-007- [T.C. Lin], 105-2410-H-002-218-MY3, 105-2811-H-002-024, 106-2811-H-002-027 [C.T. Chang]), Taiwan. L. Wang acknowledges the support from the National Natural Science Foundation (EAR-1562055). The authors thank Dr. Craig E. Martin of the University of Kansas for thoroughly editing the manuscript.

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

**Table 1.** Runoff ratio of the four watersheds during typhoon and non-typhoon periods.

| Watersheds | Typhoon period | | Non-typhoon period | |
|---|---|---|---|---|
| | Surface runoff : Precipitation | Total runoff : Precipitation | Surface runoff : Precipitation | Total runoff : Precipitation |
| A1 | 0.29 | 0.69 | 0.12 | 0.80 |
| A2 | 0.33 | 0.64 | 0.15 | 0.69 |
| F1 | 0.27 | 0.69 | 0.14 | 0.76 |
| F2 | 0.33 | 0.78 | 0.06 | 0.81 |

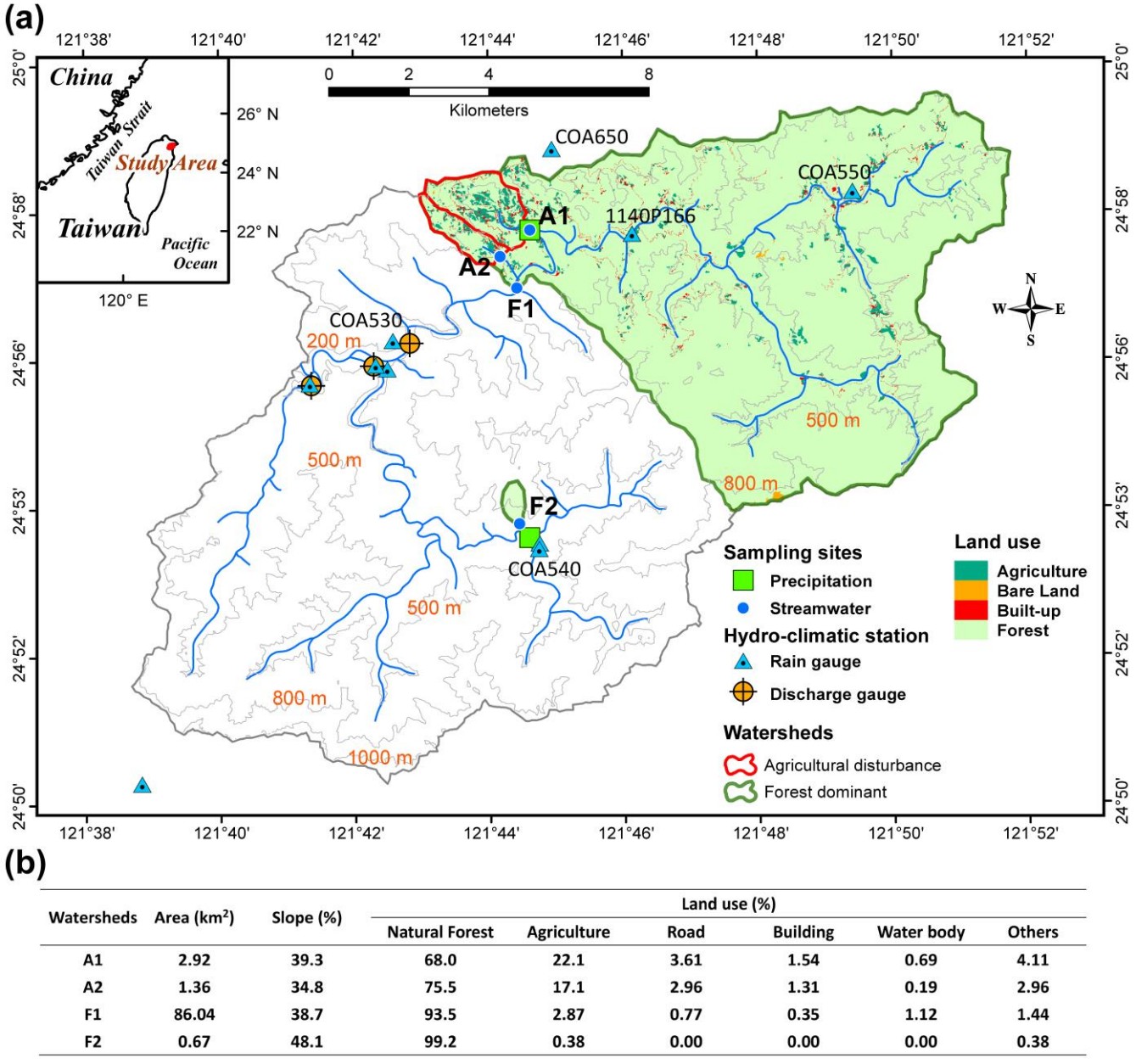

**Figure 1.** Location and land uses of the studied watersheds at the Feitsui Reservoir Watershed (a) and the basic information of four watersheds (b).

| Watersheds | Area (km²) | Slope (%) | Land use (%) | | | | | |
|---|---|---|---|---|---|---|---|---|
| | | | Natural Forest | Agriculture | Road | Building | Water body | Others |
| A1 | 2.92 | 39.3 | 68.0 | 22.1 | 3.61 | 1.54 | 0.69 | 4.11 |
| A2 | 1.36 | 34.8 | 75.5 | 17.1 | 2.96 | 1.31 | 0.19 | 2.96 |
| F1 | 86.04 | 38.7 | 93.5 | 2.87 | 0.77 | 0.35 | 1.12 | 1.44 |
| F2 | 0.67 | 48.1 | 99.2 | 0.38 | 0.00 | 0.00 | 0.00 | 0.38 |

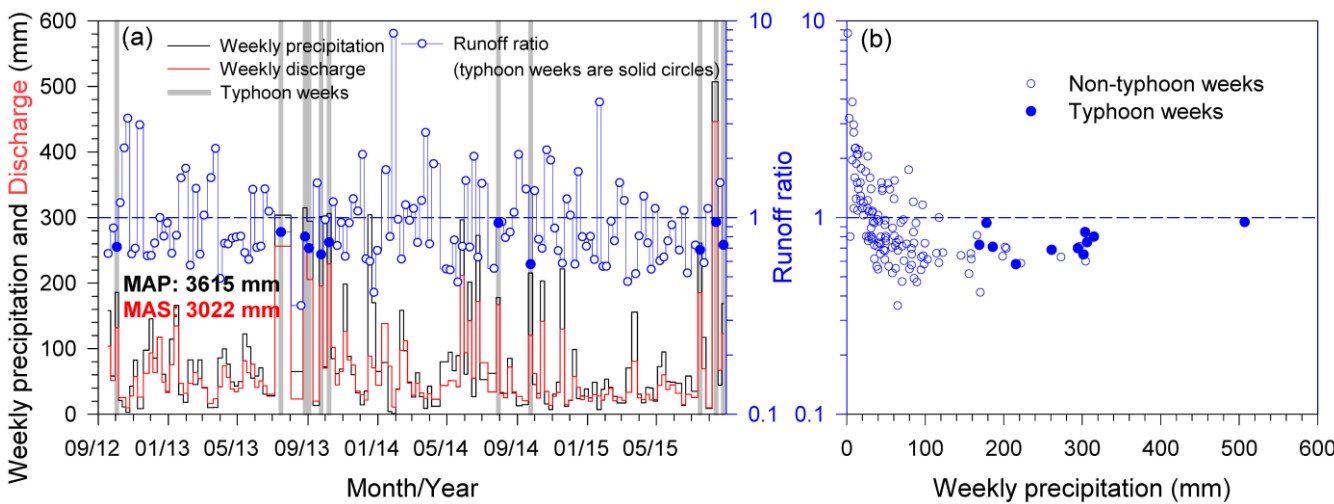

**Figure 2.** Mean weekly precipitation, discharge and runoff ratio (a), and the relationship between mean weekly precipitation and mean runoff ratio (b) of the four studied watersheds combined. MAP: Mean annual precipitation, MAS: Mean annual stream discharge.

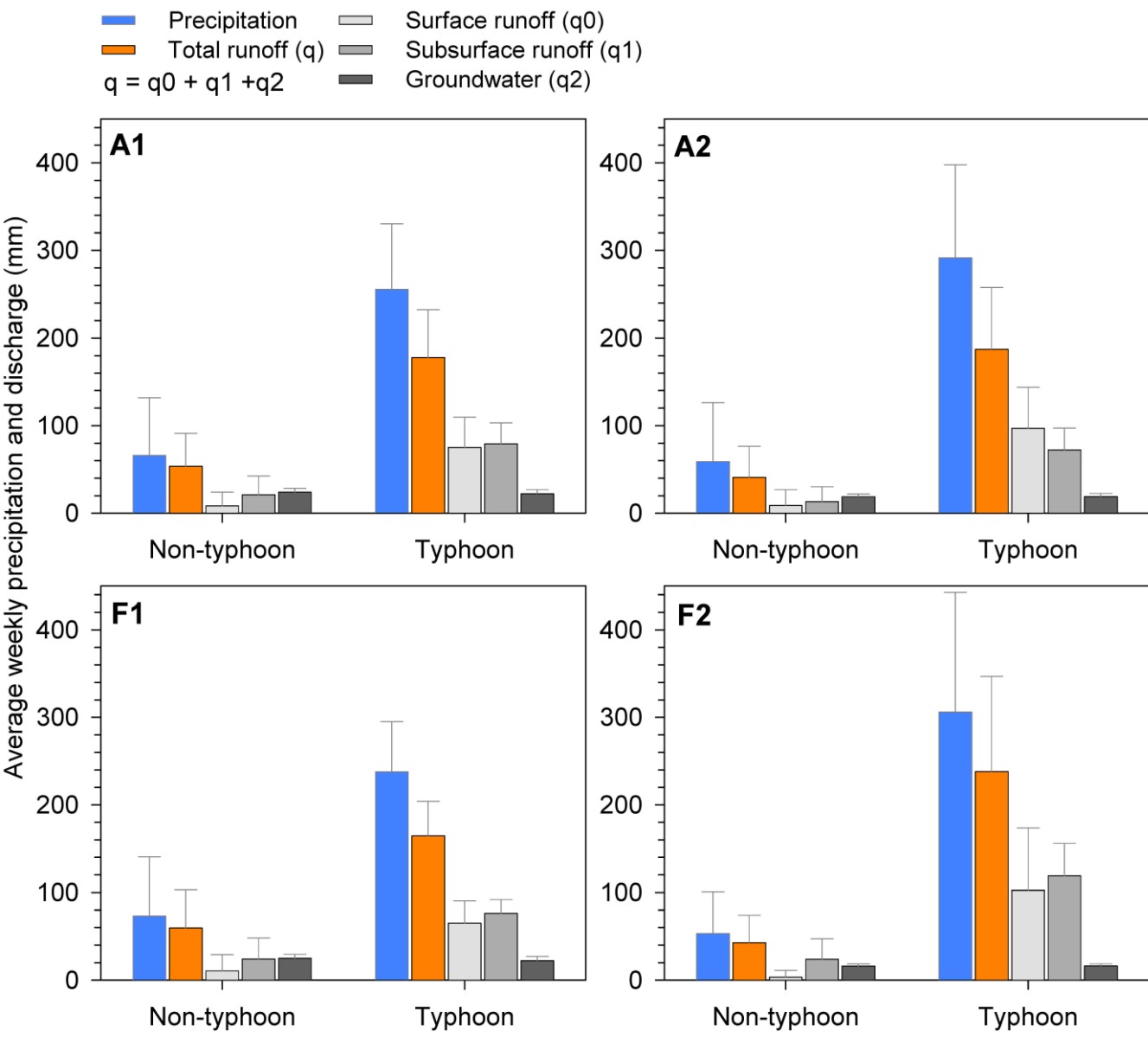

**Figure 3.** The average weekly precipitation and estimated streamflow composition during typhoon and non-typhoon period among the four watersheds. The grey bars indicate one standard deviation.

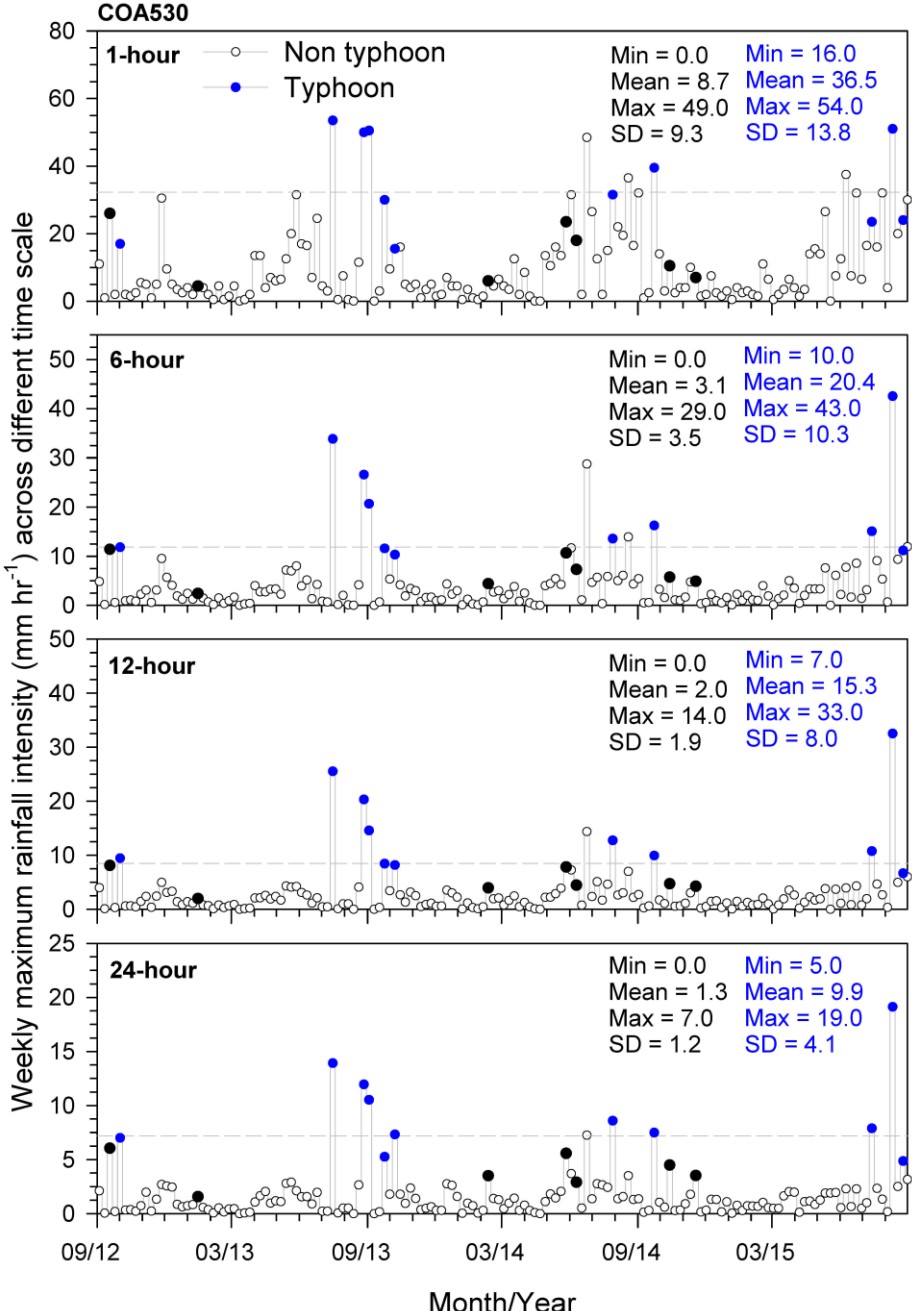

**Figure 4.** Weekly maximum 1-hr, 6-hr, 12-hr, and 24-hr precipitation of the rain gauge station COA530 (referring to the location in Fig. 1) used in this study. The black dots are non-typhoon storms with precipitation greater than that of the minimal

5 typhoon storms (160 mm).

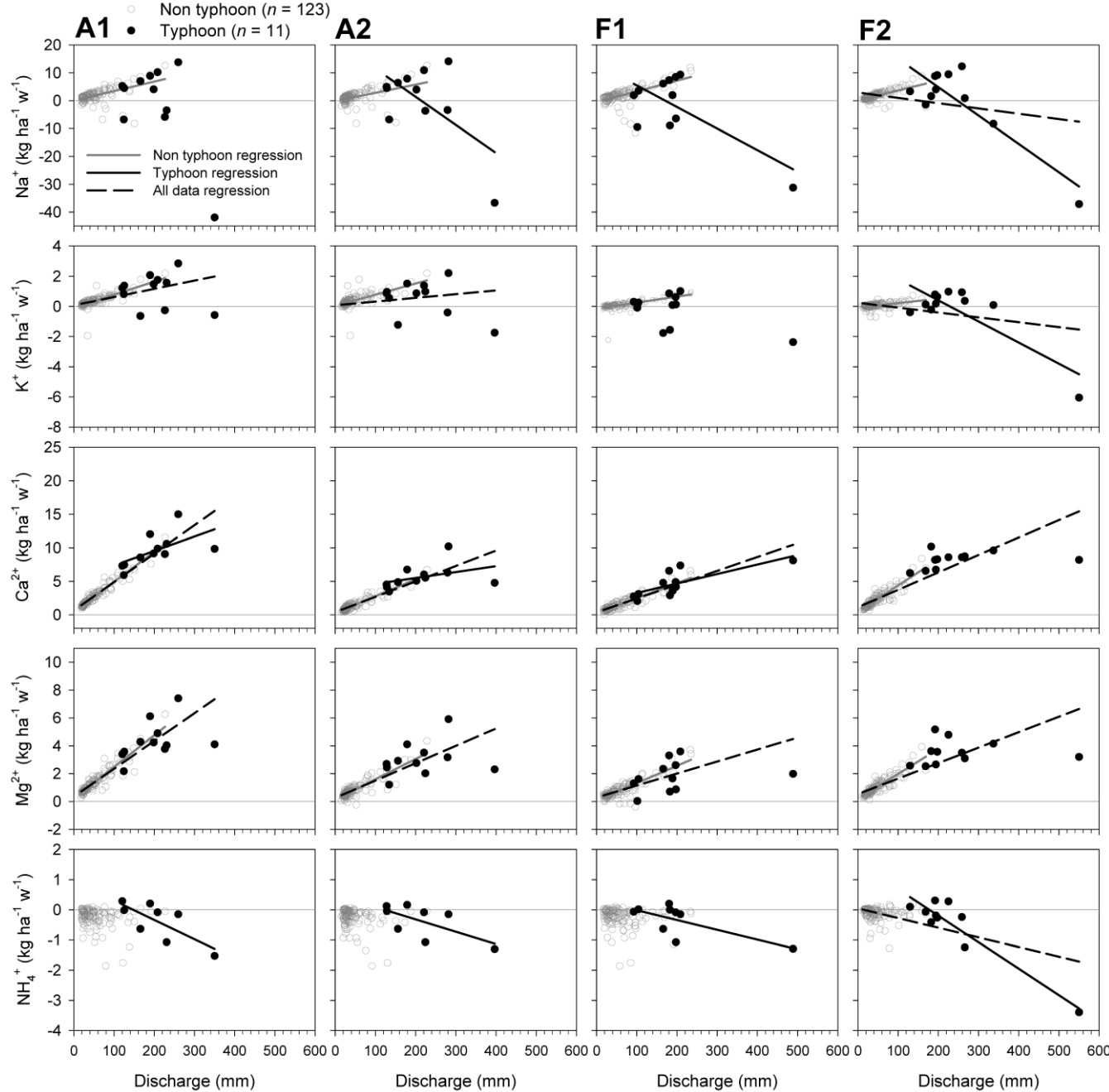

**Figure 5.** Relationship between stream discharge and nutrient budget (stream output – precipitation input) of cations ($Na^+$, $K^+$, $Ca^{2+}$, $Mg^{2+}$, and $NH_4^+$). The gray, black, and dash lines indicate significant linear regressions between discharge and ions budgets for non-typhoon, typhoon and all data, respectively. Please refer to Table S2 for the regression models and $R^2$s.

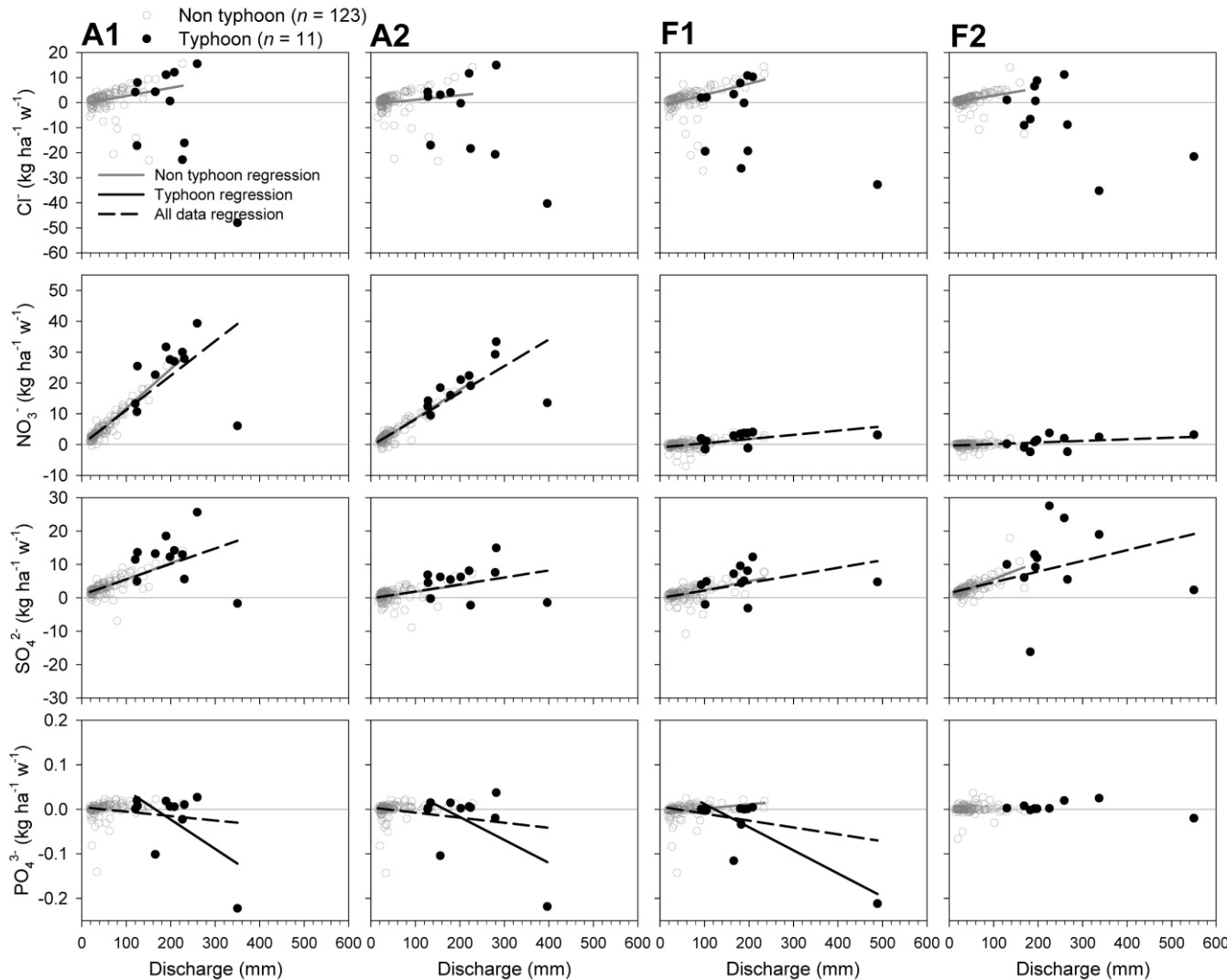

**Figure 6.** Relationship between stream discharge and nutrient budget (stream output – precipitation input) of anions ($Cl^-$, $NO_3^-$, $SO_4^{2-}$, and $PO_4^{3-}$). The gray, black, and dash lines indicate significant linear regressions between discharge and ions budgets for non-typhoon, typhoon and all data, respectively. Please refer to Table S2 for the regression models and $R^2$s.

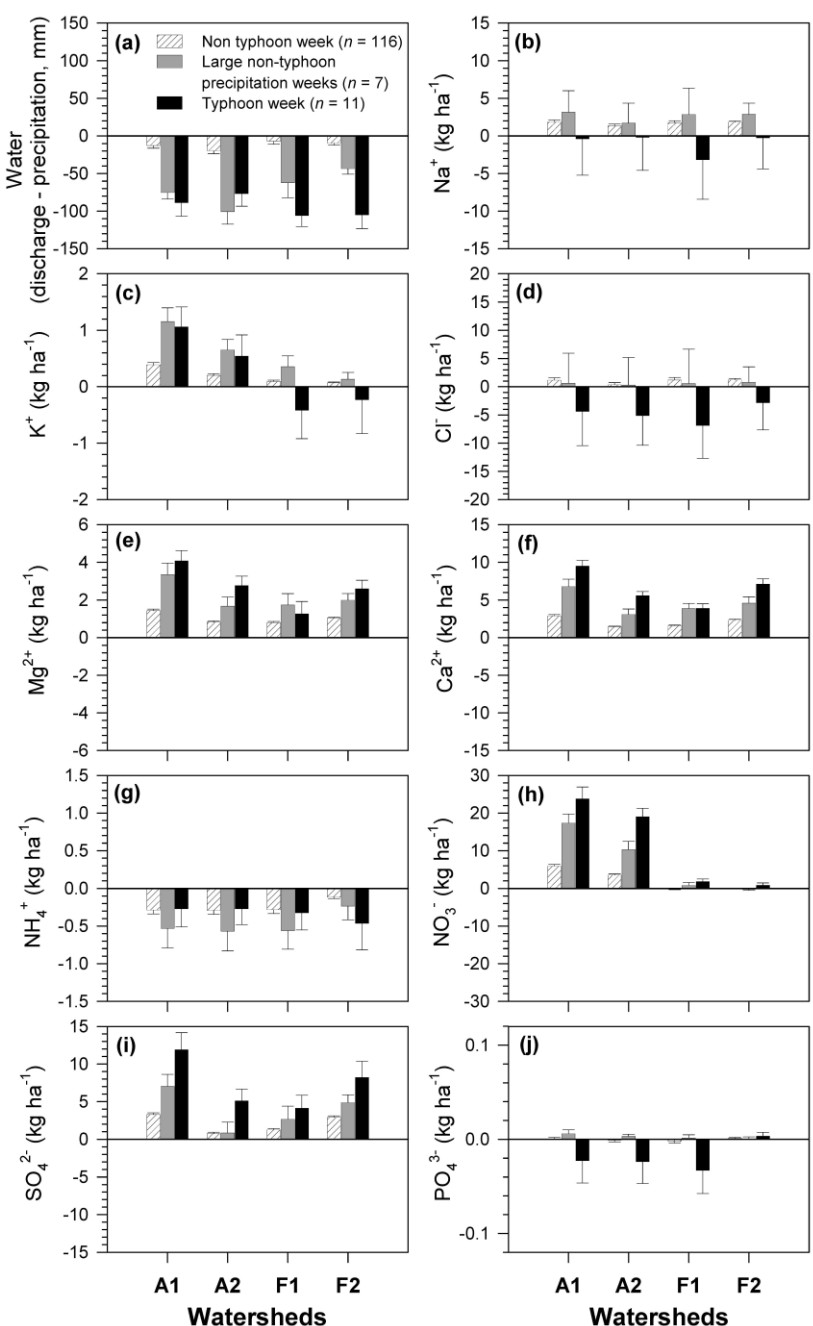

**Figure 7.** Mean weekly budget for non-typhoon weeks, large non-typhoon precipitation weeks and typhoon weeks. (a) Water quantity (stream discharge –precipitation), (b) $Na^+$, (c) $K^+$, (d) $Cl^-$, (e) $Mg^{2+}$, (f) $Ca^{2+}$, (g) $NH_4^+$, (h) $NO_3^-$, (i) $SO_4^{2-}$, and (j) $PO_4^{3-}$.