# Peer review of "Shifts in stream hydrochemistry in responses to typhoon and nontyphoon precipitation"

_Biogeosciences, 2017_

## Referee Comment (RC1) · Anonymous Referee #1 · 18 Dec 2017

Chang et al. present work from four small watersheds in northern Taiwan and report stream and precipitation hydrochemistry data over a 3-year period that encompasses 11 typhoons. It is an interesting dataset and I largely think that the methodology is adequate to answer questions related to differences between typhoon and non-typhoon hydrochemistry. There could be issues associated with using the drainage-area ratio method in watersheds with different land-uses, especially during higher flows and for watershed comparisons, but it might not matter too much for the internal hydrochemistry dynamics. The differences between the typhoon and non-typhoon hydrochemistry are striking, but also not unexpected, as storm hydrochemistry often differs from baseflow hydrochemistry. What I am missing, though, is a general discussion

WHY these pronounced differences exist (or at least an attempt at an explanation). Because of this the manuscript feels incomplete and I would not recommend publication in its current form. I would suggest the authors alter (or add to) the discussion to include possible explanations for the stark differences in typhoon and non-typhoon hydrochemistry response. Origin and fate of the water constituents should be discussed in more detail (or, for that matter, at all). Pre-typhoon conditions might matter for nutrient mobilization and typhoon runoff ratios (rather than total streamflow) could also help in interpreting the data. Are there precipitation or streamflow thresholds that change the delivery dynamics of nutrients? How might the activation of different flowpathways or water sources contribute to the differences? Please find below some more comments/suggestions.

P3 L5: Based on size, it would make sense that F1 is a 1st or 2nd order stream. However, the drainage network in Figure 1 suggests it might be a 3rd order stream.

P3 L9-10: Were the samples also analyzed within 24 hours of sample collection?

P3 L15: What is the topographic relief in the F1 watershed? Is it large enough that orographic precipitation differences should be considered?

P3 L17: The drainage-area ration method assumes similar watershed characteristics and runoff generation mechanisms. Considering that A1 and A2 have a considerable amount of agricultural area, runoff generation mechanisms are likely different, which would call into the question the comparability of streamflow volumes.

Table 1: I am not sure I understand the difference between the accumulated and total precipitation (and streamflow) values. The accumulated values are the values for just the typhoon days, while the total values are the values for the entire typhoon week? That means precipitation was measured at sub-weekly intervals and later aggregated to weekly values? This is not immediately clear from paragraph 2.3.

P4 L4-18: It would be important to have more basic hydrologic data in this paragraph.
For starters, I am missing streamflow hydrographs for the study periods. For a better overview of the general hydrologic conditions. This paragraph and Table 1 contain data to calculate typhoon runoff ratios (the amount of precipitation that becomes runoff over a period of time: streamflow/precipitation), but it would also be interesting to see annual runoff ratios for the watersheds with and without the typhoon periods. Assessing pre-typhoon conditions might also be helpful for interpreting the different response between typhoon and non-typhoon periods but also the variability within the typhoon responses.

P4 L17 and Figure 2: Record over what time? What are the "5, 8, 9, and 9 typhoon weeks" and how are they shown in Figure 2? What do the dashed lines in Figure 2 represent?

P4 L14-18: This whole paragraph is about hourly intensities, not precipitation totals.

Table 2: It's good to list the regression models but this could be supplemental information in my opinion.

—————————————————————

---

## Referee Comment (RC2) · Anonymous Referee #2 · 10 Jan 2018

This manuscript presents relationships between water budget (precipitation and discharge) and weekly hydrochemical data in four watersheds within the Feitsui Reservoir Watershed (FRW) in northern Taiwan. The dataset spans three years and encompasses eleven Typhoon systems. The authors use these data to draw inference about distinct hydrochemical response during typhoon vs. non-typhoon times, both in terms of variability and direction. Additionally, the watersheds differ in size and relative proportion of agricultural land - in this case largely heavily fertilized tea plantations - and they use this difference to examine the effect of land use change on response to typhoon events.

The authors have an interesting and appropriate dataset to address their questions, and additionally seem to have chosen an ideal location to elucidate dependence of hydrochemical response on storm intensity. Their methods are logical and results indicate striking differences between watershed response during typhoon and non-typhoon times.

I have two suggestions for review I would characterize as "major," along with multiple minor suggestions and comments, which are delineated below. Given response to these suggestions, this paper seems like a good candidate for publication in Biogeosciences.

1. Discharge estimation and enhanced hydrologic analysis: The author's use of the "area-ratio" method of estimating discharge is far from ideal, although it seems to be unavoidable in the absence of other gauging stations within the study watersheds. It is surprising they did not at least perform weekly in-situ measurements of discharge (area-velocity or dilution gauging) in conjunction with their chemistry sampling. Additionally, they present no hydrography or additional hydrologic analyses, for example runoff ratio at annual and storm scales, which may help interpret their weekly data and results.

1a. I would like to see a more thorough explanation of both their method for estimating discharge (rather than requiring the reader to reference to one of the author's previous publications) and a discussion of the method's limitations. This should not be long or intensive, but should be sufficient to help the reader understand the reasons for doing so and the potential effects on the analysis.

1b. The author's should strongly consider including standard hydrographs coupled with their hyetographs for context, and potentially also include annual and a storm-by-storm analysis of runoff coefficients. The latter may require some time, but would be very valuable in interpreting their dataset from the standpoint of variable runoff generation processes due to intensity. Additional context such as intensity-duration curves with

differentially colored typhoon and non-typhoon events would also help make points they make primarily in text.

2. Process-based discussion: With the richness of their chemical dataset, I expected the discussion to be an ideal opportunity to discuss the observed non-linearities in hydrochemical response in the context of physical and biogeochemical process. The authors rightly make no claim at their ability to rigorously distinguish the physical or biogeochemical processes responsible for the dynamics the observe, but there is a rich literature within watershed hydrology and biogeochemistry addressing each of these chemical constituents which would appropriately be used to contextualize their findings. What do the nutrient dynamics suggest about stream response to extreme events? Does stream productivity or nutrient saturation factor in to the differences between A and F watersheds? Does the response of weathering products suggest activation of distinct flowpaths in non-typhoon vs typhoon events? Any of these or more would be appropriate discussion points and would help the reader move past observation to interpretation.

See below for minor suggestions and comments delineated by page and line number:

P2 L25-37: More descriptive climatic information would help the portion of the audience not familiar with the location: annual precipitation, climate zone, seasonality, etc. L30-31 could include actual quantification.

P3 L3-6: This watershed description could go more appropriately in the previous section as study area description.

P3 L7: At what frequency? Or was it recorded continuously?

P3 L9-10: Could the authors explain more about how they handled their samples, particularly with regard to the nutrients? As described their methods may be problematic with regard to nutrients. Particularly in what I would assume is a relatively warm, high-productivity system, samples should be field-filtered or chemically stabilized and then

frozen as soon as possible. Alternatively they may be analyzed immediately upon re-turn from the field. If they did neither, it might be appropriate to include some discussion of the effect of uptake on their samples. For example, in the lower NO3, F watersheds there may have been more relative uptake after sampling than in the presumably eu-trophic, potentially saturated, A watersheds. NH4 is also challenging because it is so readily nitrified.

P3 L15-20: More is needed here. As mentioned above, I'd like to see a better descrip-tion and justification for the discharge scaling method they used.

P3 Section 2.4: Here the authors spend significant time discussing another potential source of error; something similar is needed for discharge.

P4 L20-21: This seems like an interesting finding, rather than differences in regression direction, since many of the regressions for typhoon periods have very low predictabil-ity.

P4-5 Section 3.3: Reiterating above, the change in direction of the regression is less compelling to me than the dramatic change in spread of the point cloud between non-typhoon and typhoon periods (i.e., much lower predictability).

P6 L9-10: I'd be interested in a comparison of extreme, non-typhoon events with com-parable typhoon events. Do the authors feel that it is merely the high intensity of the ty-phoons that cause the unique hydrochemical response? Or is there something unique about typhoons? High winds or variable winds that can damage forests and farms that change delivery of ions to streams?

P7 L8-10: To reiterate a point from the methods, I would encourage the authors to think through the potential effect of nutrient uptake after sampling, if indeed there was no fil-tering or other stabilization and there was a period of multiple days before analyzing the samples. If the F and A watersheds have distinct nutrient regimes, it is also rea-sonable to expect they may exhibit distinct uptake responses, which could differentially

affect each set of samples.

P7-8 Section 4.3: The authors provide some interesting, process-based context for their findings in this section! This is the type of discussion I would like to see throughout with respect to all of their findings.

Tables 1 and 2: These could go in a supplementary section

Figures 2 and 3: The authors did a good job of making their figures interpretable in grayscale by choosing their colors and using open vs closed circles. They could further improve this by doing something similar for the regression lines shown here, perhaps dotted lines for the black, total dataset regressions? Purple and blue lines should be distinguishable in grayscale.

---

## Author Comment (AC2) · 27 Jan 2018

Dear editor and reviewer,

We have combined all comments and responses into a single PDF document. Please check the supplement (attachment).

Best regards,

Teng-Chiu Lin, Department of Life Science, National Taiwan Normal University, #88, Section 4, DingChow Road, Taipei, 11677, Taiwan

Tel: 886-2-77346240

[Figure]

Fax: 886-2-29312904

Please also note the supplement to this comment:
https://www.biogeosciences-discuss.net/bg-2017-394/bg-2017-394-AC2-
supplement.pdf
* * *
[Figure]

**Supplement:**

The black and blue texts are comments from reviewers and author's responses respectively.

**Comments from Reviewer #2,**

This manuscript presents relationships between water budget (precipitation and discharge) and weekly hydrochemical data in four watersheds within the Feitsui Reservoir Watershed (FRW) in northern Taiwan. The dataset spans three years and encompasses eleven Typhoon systems. The authors use these data to draw inference about distinct hydrochemical response during typhoon vs. non-typhoon times, both in terms of variability and direction. Additionally, the watersheds differ in size and relative proportion of agricultural land - in this case largely heavily fertilized tea plantations - and they use this difference to examine the effect of land use change on response to typhoon events. The authors have an interesting and appropriate dataset to address their questions. and additionally seem to have chosen an ideal location to elucidate dependence of hydrochemical response on storm intensity. Their methods are logical and results indicate striking differences between watershed response during typhoon and non-typhoon times. I have two suggestions for review I would characterize as "major," along with multiple minor suggestions and comments, which are delineated below. Given response to these suggestions, this paper seems like a good candidate for publication in Biogeosciences. 1. Discharge estimation and enhanced hydrologic analysis: The author's use of the "arearatio" method of estimating discharge is far from ideal, although it seems to be unavoidable in the absence of other gauging stations within the study watersheds. It is surprising they did not at least perform weekly in-situ measurements of discharge (area-velocity or dilution gauging) in conjunction with their chemistry sampling. Additionally, they present no hydrography or additional hydrologic analyses, for example runoff ratio at annual and storm scales, which may help interpret their weekly data and results.

1a. I would like to see a more thorough explanation of both their method for estimating discharge (rather than requiring the reader to reference to one of the author's previous publications) and a discussion of the method's limitations. This should not be long or intensive, but should be sufficient to help the reader understand the reasons for doing so and the potential effects on the analysis.

Reply: We agree that the area ratio method is not ideal and re-estimated stream discharge using the Hydrologiska Byråns Vattenbalansavdelning (HBV) model. We updated all the figures and tables based on the new calculations. The basic patterns do not change but some details are different. The following detail is added to the Materials and Methods section.

"Stream discharge of the four ungauged watersheds was also simulated by the HBV model processed through TUWmodel (ver. 0.1-8) (Paraika et al., 2013). Five daily rain gauges. maintained by Water Resource Agency (WRA), and five metrological stations, maintained by the Central Weather Bureau of Taiwan (CWB) with hourly observed rainfall, temperature, wind speed, and solar radiation were used to estimate daily rainfall and potential evapotranspiration. The daily evapotranspiration is also observed by Taipei Feitsui Reservoir Administration (TFRA, Taiwan) at the Feitsui meteorological station. The observed rainfall, temperature and evapotranspiration were applied into 20 sub-catchments with Thiessen polygon method. Daily discharge was monitored in three main tributaries of Baishi Creek by TFRA. In the calibration against the observed values, parameters were generated by the package DEoptim (ver. 2.2-4) (Mullen et al., 2011). Three objective functions, Nash Efficient Coefficient (NSE), its power of 2 and log scale, were used to adjust the model to suit normal, extreme, and low flow conditions. The validation gauge is located in the inflow of dam of reservoir. The modelled daily discharge was aggregated into weekly discharge." We updated all the figures and tables based on the new calculations. The basic patterns do not change but some details are different. The cited references are listed below.

- Parajka, J., Viglone, A., Salinas, R. M., Sviapalan, M., and Blöschl, G.: Comparative assessment of predictions in ungauged basins Part 1: Runoff-hydrograph studies, Hydrol. Earth Sys. Sci., 17, 1783–1795, doi:10.5194/hess-17-1783-2013, 2013.
- Mullen, K. M., Ardia, D., Gil, D. L., Windover, D., and Cline, J.: "DEoptim: An R Package for Global Optimization by Differential Evolution, J. Stat. Softw., 40, 1–26, doi: 10.18637/jss.v040.i06, 2011.

1b. The author's should strongly consider including standard hydrographs coupled with their hyetographs for context, and potentially also include annual and a storm-by-storm analysis of runoff coefficients. The latter may require some time, but would be very valuable in interpreting their dataset from the standpoint of variable runoff generation processes due to intensity. Additional context such as intensity-duration curves with differentially colored typhoon and non-typhoon events would also help make points they make primarily in text. Reply: We added standard hydrographs to the results and calculated weekly runoff ratios and examined their relationship to precipitation. The following information is added to the revision. "During the sampling period, weekly precipitation ranged from 1 mm to 470 mm while weekly streamflow ranged from 10 mm to 446 mm (Fig. 2)". We also calculated weekly runoff ratio and the mean runoff ratio for typhoon period and non-typhoon period. In the Results section, we added "The weekly runoff ratio was negatively related to precipitation quantity and was highly variable during the non-typhoon period but varied much less during the typhoon period (Fig. 2)".

Figure 2. Mean weekly precipitation, discharge and runoff (a), and the relationship between mean weekly precipitation and mean runoff ratio (b) of the four studied watersheds combined. MAP: Mean annual precipitation, MAS: Mean annual stream discharge.

2. Process-based discussion: With the richness of their chemical dataset, I expected the discussion to be an ideal opportunity to discuss the observed non-linearities in hydrochemical response in the context of physical and biogeochemical process. The authors rightly make no claim at their ability to rigorously distinguish the physical or biogeochemical processes responsible for the dynamics the observe, but there is a rich literature within watershed hydrology and biogeochemistry addressing each of these chemical constituents which would appropriately be used to contextualize their findings. What do the nutrient dynamics suggest about stream response to extreme events? Does stream productivity or nutrient saturation factor in to the differences between A and F watersheds? Does the response of weathering products suggest activation of distinct flowpaths in non-typhoon vs typhoon events? Any of these or more would be appropriate discussion points and would help the reader move past observation to interpretation. Reply: We substantially extended our discussion to include possible explanations to the differences in the patterns between typhoon and non-typhoon periods. The following is added to the discussion. In the Discussion section, we added "Stream discharge originates from three sources, surface runoff, subsurface runoff and groundwater discharge. Among the three sources, groundwater discharge was more important during low than high flow periods, whereas the contribution from surface runoff should be more important during heavy storms than small storms. The contribution from subsurface flow probably dominated the discharge at our study site, especially in F1 and F2 because a study at a natural forest

12 km Southeast from our study site indicated that even during a heavy typhoon storm, with precipitation near 700 mm in two days, there was no observable surface runoff (Lin et al., 2011). The contribution from groundwater and subsurface runoff to total discharge likely resulted in the very high runoff ratios for weeks with small amount of precipitation. For example, in 28 January 2014, the weekly precipitation and discharge were 1.5 mm and 13 mm, respectively, which led to the highest runoff ratio, 8.7, for the entire study period (Fig. 2). Groundwater is enriched with ions derived from rock weathering such as  $K^+$ . Ca2+, and  $Mq^{2+}$ . In addition, pre-storm subsurface runoff has a longer contact time with soils that are also rich in these cations and  $SO_{4^2}$ . Thus, the greater contributions from groundwater and subsurface runoff in the non-typhoon period likely contributed to the greater (positive) slopes between discharge and flux of these ions for the non-typhoon period than typhoon period (Figs. 4 and 5). The second possible reason for the greater slopes between discharge and fluxes of many ions during the non-typhoon period is the differences in ion concentration between typhoon and non-typhoon storms. Clear sky characteristic of the one or two days before a typhoon is typical because the outskirt air masses of the typhoon "blow" away most air pollutants. As a result, precipitation associated with typhoons have low concentrations of ions with terrestrial sources (Lin et al., 2011). In our study, mean concentrations of all ions were lower during typhoon period than non-typhoon period (Table S3) and this diluted precipitation ion concentrations overrode quantity effect and contributed to the smaller increase in ion flux with increasing discharge (Figs. 4 and 5)."

Lin, T. C., Hamburg, S. P., Lin, K. C., Wang, L. J., Chang, C. T., Hsia, Y. J., Vadeboncoeur, M. A., McMullen, C. M. C., and Liu, C. P.: Typhoon disturbance and forest dynamics: lessons from a northwest Pacific subtropical forest, Ecosystems, 14, 127–143, doi: 10.1007/s10021-010-9399-1, 2011.

Table S3. The mean (one standard deviation) concentrations of ions (mg/l) in precipitation during non-typhoon (Non\_Ty) and typhoon periods

|         | H+          | Na + | K 2+ | Ca 2+ | Mg 2+ | NH4 + | Cl-         | NO3 - | SO42-       | PO4 3- |
|---------|-------------|-----------------|-----------------|------------------|------------------|------------------|-------------|------------------|-------------|-------------------|
| A1      |             |                 |                 |                  |                  |                  |             |                  |             |                   |
| Non_Ty  | 0.06 (0.06) | 4.15 (8.35)     | 0.64 (1.36)     | 0.84 (1.36)      | 0.59 (1.04)      | 0.85 (1.35)      | 9.43 (22.9) | 3.97 (6.27)      | 5.18 (7.26) | 0.04 (0.15)       |
| Typhoon | 0.02 (0.02) | 3.78 (3.51)     | 0.43 (0.45)     | 0.48 (0.36)      | 0.51 (0.41)      | 0.12 (0.14)      | 6.05 (5.21) | 0.40 (0.70)      | 1.54 (1.57) | 0.01 (0.02)       |
| F2      |             |                 |                 |                  |                  |                  |             |                  |             |                   |
| Non_Ty  | 0.04 (0.06) | 3.37 (8.17)     | 0.61 (1.66)     | 0.84 (1.91)      | 0.55 (1.08)      | 0.55 (1.18)      | 7.96 (23.9) | 2.56 (9.10)      | 4.84 (10.2) | 0.01 (0.05)       |
| Typhoon | 0.01 (0.02) | 3.36 (2.75)     | 0.37 (0.39)     | 0.32 (0.23)      | 0.46 (0.36)      | 0.14 (0.17)      | 5.03 (3.96) | 0.39 (0.53)      | 2.11 (2.80) | 0.002 (0.003)     |

---

## Author Response (AR1)

The black and blue texts are comments from reviewers and author's responses respectively.

5    --------------------------------
**Comments from Reviewer #1,**

Chang et al. present work from four small watersheds in northern Taiwan and report stream and precipitation hydrochemistry data over a 3-year period that encompasses 11 typhoons.

10   It is an interesting dataset and I largely think that the methodology is adequate to answer questions related to differences between typhoon and non-typhoon hydrochemistry. There could be issues associated with using the drainage-area ratio method in watersheds with different land-uses, especially during higher flows and for watershed comparisons, but it might not matter too much for the internal hydrochemistry dynamics. The differences

15   between the typhoon and non-typhoon hydrochemistry are striking, but also not unexpected, as storm hydrochemistry often differs from baseflow hydrochemistry. What I am missing, though, is a general discussion WHY these pronounced differences exist (or at least an attempt at an explanation). Because of this the manuscript feels incomplete and I would not recommend publication in its current form. I would suggest the authors alter (or

20   add to) the discussion to include possible explanations for the stark differences in typhoon and non-typhoon hydrochemistry response. Origin and fate of the water constituents should be discussed in more detail (or, for that matter, at all). Pre-typhoon conditions might matter for nutrient mobilization and typhoon runoff ratios (rather than total streamflow) could also help in interpreting the data. Are there precipitation or streamflow thresholds that change

25   the delivery dynamics of nutrients? How might the activation of different flowpathways or water sources contribute to the differences? Please find below some more comments/suggestions.
Reply: We substantially expanded our discussion to include explanations to the observed striking differences between typhoon and non-typhoon periods. Specially, we added the

30   differences in runoff ratio between the two periods in a new Figure 2. The added discussion is listed below.
*"4.1    Differences between typhoon and non-typhoon periods*
*The striking differences in the discharge-budget patterns between typhoon and non-typhoon periods should be related to changes in the relative proportion of sources of stream*

35   *discharge. Stream discharge originates from three sources, surface runoff, subsurface runoff and groundwater. Among the three sources, groundwater was more important during low than high flow periods, whereas the contribution from surface runoff should be more important during heavy storms than small storms. The contribution from subsurface flow*

*probably dominated the discharge at our study site, especially in F1 and F2 because a study at a natural forest 12 km Southeast from our study site indicated that even during a heavy typhoon storm, with precipitation near 700 mm in two days, there was no observable surface runoff (Lin et al., 2011). The contribution from groundwater and subsurface runoff to total discharge likely resulted in the very high runoff ratios for weeks with small amount of precipitation. For example, in 28 January 2014, the weekly precipitation and discharge were 1.5 mm and 13 mm, respectively, which led to the highest runoff ratio, 8.7, for the entire study period (Fig. 2).*

*Changes in relatively contributions of different sources of water (or old water relative to new water) on stream discharge, play a key role in regulating ion concentrations during a storm and between periods of different flow rates (Elwood and Turner, 1989; Giusti and Neal 1993; Bishop et al., 2004). Among the three sources, groundwater is enriched with ions derived from rock weathering such as $K^+$, $Ca^{2+}$, and $Mg^{2+}$, pre-storm subsurface runoff have a longer contact time with soils that are also rich in these cations and $SO_4^{2-}$, and surface runoff largely reflect precipitation chemistry. A study of storm solute transport in a forested watershed in northern Taiwan, 12 km Southwest of our study site, indicated that concentrations of $Na^+$, $Ca^{2+}$, $Mg^{2+}$, $Cl^-$, and $SO_4^{2-}$ were diluted due to the mixing of large quantities whereas concentrations of $K^+$, $NH_4^+$ and $NO_3^-$ were enhanced during high flows (Wang et al. 1996). In our study, the greater contributions from groundwater and subsurface runoff in the non-typhoon period likely contributed to the greater (more positive) slopes between discharge and budget of many ions for the non-typhoon period than typhoon period, in which many of the relationships were not significant (Figs. 4 and 5). The second possible reason for the greater slopes between discharge and budget of many ions during the non-typhoon period is the differences in ion concentration between typhoon and non-typhoon storms. The day or two days before a typhoon typically has clear sky because the outskirt air masses of the typhoon "blow" away most air pollutants. As a result, precipitation associated with typhoons have low concentrations of ions with terrestrial sources (Lin et al., 2011). In our study, mean concentrations of all ions were lower during typhoon period than non-typhoon period (Table S3) and this diluted precipitation ion concentrations overrode quantity effect and contributed to the smaller increases in budget with increasing discharge in the typhoon period than the non-typhoon period (Figs. 4 and 5)."*

[Figure]

Figure 2: Mean weekly precipitation, discharge and runoff (a), and the relationship between mean weekly precipitation and mean runoff ratio (b) of the four studied watersheds combined. MAP: Mean annual precipitation, MAS: mean annual stream discharge.

**Table S3**. The mean (one standard deviation) concentrations of ions (mg/l) in precipitation during non-typhoon and typhoon periods.

| | H | Na$^+$ | K$^+$ | Ca$^{2+}$ | Mg$^{2+}$ | NH$_4^+$ | Cl$^-$ | NO$_3^-$ | SO$_4^{2-}$ | PO$_4^{3-}$ |
|---|---|---|---|---|---|---|---|---|---|---|
| **A1** | | | | | | | | | | |
| Non-typhoon | 0.06 (0.06) | 4.15 (8.35) | 0.64 (1.36) | 0.84 (1.36) | 0.59 (1.04) | 0.85 (1.35) | 9.43 (22.9) | 3.97 (6.27) | 5.18 (7.26) | 0.04 (0.15) |
| Typhoon | 0.02 (0.02) | 3.78 (3.51) | 0.43 (0.45) | 0.48 (0.36) | 0.51 (0.41) | 0.12 (0.14) | 6.05 (5.21) | 0.40 (0.70) | 1.54 (1.57) | 0.01 (0.02) |
| **F2** | | | | | | | | | | |
| Non-typhoon | 0.04 (0.06) | 3.37 (8.17) | 0.61 (1.66) | 0.84 (1.91) | 0.55 (1.08) | 0.55 (1.18) | 7.96 (23.9) | 2.56 (9.10) | 4.84 (10.2) | 0.01 (0.05) |
| Typhoon | 0.01 (0.02) | 3.36 (2.75) | 0.37 (0.39) | 0.32 (0.23) | 0.46 (0.36) | 0.14 (0.17) | 5.03 (3.96) | 0.39 (0.53) | 2.11 (2.80) | 0.002 (0.003) |

[Figure]

Figure 4: Relationship between stream discharge and nutrient budget (stream output – precipitation input) of cations ($Na^+$, $K^+$, $Ca^{2+}$, $Mg^{2+}$, and $NH_4^+$). The gray, black, and dash lines indicate significant linear regressions between discharge and ions budgets for non-typhoon, typhoon and all data, respectively. Please refer to Table S2 for the regression models and $R^2$s.

[Figure]

Figure 5: Relationship between stream discharge and nutrient budget (stream output – precipitation input) of anions ($Cl^-$, $NO_3^-$, $SO_4^{2-}$, and $PO_4^{3-}$). The gray, black, and dash lines indicate significant linear regressions between discharge and ions budgets for non-typhoon, typhoon and all data, respectively. Please refer to Table S2 for the regression models and $R^2$s.

1. P3 L5: Based on size, it would make sense that F1 is a 1st or 2nd order stream. However, the drainage network in Figure 1 suggests it might be a 3rd order stream.
Reply: Watershed F1 is indeed a 3rd order stream. It is not uncommon to have 3rd order streams with an area smaller than 100 km$^2$ in Taiwan due to the abundant precipitation and rough topography (so that many upstream watersheds are small).

2. P3 L9-10: Were the samples also analyzed within 24 hours of sample collection?
Reply: We filtered the samples the same day of collection and the chemical analysis were mostly completed within two weeks. We added one sentence to deliver this information.
*"After the measurement of pH and conductivity, samples were filtered (0.45 µm filter paper) mostly within eight hours of sample collection."*

3. P3 L15: What is the topographic relief in the F1 watershed? Is it large enough that orographic precipitation differences should be considered?
Reply: F1 has a mean slope steepness of 38.7% so that there could be some orographic precipitation differences. In response to this comment, we used 10 rainfall stations (instead of two in the original manuscript) to simulate the discharges of the four sites via the Hydrologiska Byråns Vattenbalansavdelning (HBV) model. The site map has been changed to include the locations of the stations. The areal rainfall from Thiessen polygon was applied, and thus the rainfall spatial heterogeneity has been considered partially. For orographic precipitation differences in F1, we conducted pair comparisons among the three rainfall stations (C0A550, 1140P166, and C0A650, outside the F1). The result showed that the slopes were close to 1.0 and the $R^2$ values were greater than 0.65. Based on the result, the differences among the three stations are less than 10% on a weekly basis. This demonstrated that the orographic effect exists and the differences in weekly precipitation among stations could be as large as 100 mm. However, the high correlations indicated that using Thiessen polygon interpolation is valid for representing the rainfall amount in F1. We added the following to the revision.

*"Precipitation in mountainous area is quite dynamic due to the interaction between orography and circulation. Following Huang et al. (2011), we used 10 rainfall stations to simulate the discharges of the four sites via Hydrologiska Byråns Vattenbalansavdelning (HBV) model. The areal rainfall from Thiessen polygon was applied, and thus the rainfall spatial heterogeneity has been considered partially. Precipitation of each of the four watersheds was then obtained from the spatial distribution of precipitation."*

[Figure]

| Watersheds | Area (km²) | Slope (%) | Land use (%) | | | | | |
|---|---|---|---|---|---|---|---|---|
| | | | Natural Forest | Agriculture | Road | Building | Water body | Others |
| A1 | 2.92 | 39.3 | 68.0 | 22.1 | 3.61 | 1.54 | 0.69 | 4.11 |
| A2 | 1.36 | 34.8 | 75.5 | 17.1 | 2.96 | 1.31 | 0.19 | 2.96 |
| F1 | 86.04 | 38.7 | 93.5 | 2.87 | 0.77 | 0.35 | 1.12 | 1.44 |
| F2 | 0.67 | 48.1 | 99.2 | 0.38 | 0.00 | 0.00 | 0.00 | 0.38 |

*Figure 1: Location and land uses of the studied watersheds at the Feitsui Reservoir Watershed (a) and the basic information of four watersheds (b).*

[Figure]

Relationship of weekly precipitation among three gauge stations of the study site.

5    Huang, J. C., Kao, S. J., Lin, C. Y., Chang, P. L., Lee, T. Y., and Li, M. H.: Effect of
        subsampling tropical cyclone rainfall on flood hydrograph response in a subtropical
        mountainous catchment, J. Hydrol., 409, 248–261, doi:10.1016/j.jhydrol.2011.08.037,
        2011.

10   4.   P3 L17: The drainage-area ration method assumes similar watershed characteristics
         and runoff generation mechanisms. Considering that A1 and A2 have a considerable
         amount of agricultural area, runoff generation mechanisms are likely different, which
         would call into the question the comparability of streamflow volumes.
     Reply: Following this comment and the general comment, we re-calculated stream
15   discharge using the Hydrologiska Byråns Vattenbalansavdelning (HBV) model. The
     following detail is added to the Materials and Methods.
     *"Stream discharge of the four ungauged watersheds was also simulated by the HBV model
     processed through TUWmodel (ver. 0.1-8) (Parajka et al., 2013). Five daily rain gauges,
     maintained by Water Resource Agency (WRA), and five metrological stations, maintained*
20   *by the Central Weather Bureau (CWB) of Taiwan with hourly observed rainfall, temperature,
     wind speed, and solar radiation were used to estimate daily rainfall and potential
     evapotranspiration. The daily evapotranspiration is also observed by Taipei Feitsui
     Reservoir Administration (TFRA, Taiwan) at the Feitsui meteorological station. The*

*observed rainfall, temperature and evapotranspiration were applied into 20 sub-catchments with Thiessen polygon method. Daily discharge was monitored in three main tributaries of Baishi Creek by TFRA. In the calibration against the observed values, parameters were generated by the package DEoptim (ver. 2.2-4) (Mullen et al., 2011). Three objective*
5 *functions, Nash Efficient Coefficient (NSE), its power of 2 and log scale, were used to adjust the model to suit normal, extreme, and low flow conditions. The validation gauge is located in the inflow of dam of reservoir. The modelled daily discharge was aggregated into weekly discharge."* We updated all the figures and tables based on the new calculations. The basic patterns do not change but some details are different. The cited references are
10 listed below.

Parajka, J., Viglone, A., Salinas, R. M., Sviapalan, M., and Blöschl, G.: Comparative assessment of predictions in ungauged basins - Part 1: Runoff-hydrograph studies, Hydrol. Earth Sys. Sci., 17, 1783–1795, doi:10.5194/hess-17-1783-2013, 2013.
Mullen, K. M., Ardia, D., Gil, D. L., Windover, D., and Cline, J.: "DEoptim: An R Package for
15 Global Optimization by Differential Evolution, J. Stat. Softw., 40, 1–26, doi: 10.18637/jss.v040.i06, 2011.

Table 1: I am not sure I understand the difference between the accumulated and total precipitation (and streamflow) values. The accumulated values are the values for just the
20 typhoon days, while the total values are the values for the entire typhoon week? That means precipitation was measured at sub-weekly intervals and later aggregated to weekly values? This is not immediately clear from paragraph 2.3.

Reply: We agree that the current expression is a bit confusing. We changed *"Accumulated prec. of specific typhoon"* to *"Precipitation between the first and last typhoon warnings"* and
25 *changed "Accumulated discharge of specific typhoon""* to *"Discharge between the first and last typhoon warnings". We think this should make it clear that the numbers in the two columns represent precipitation contributed by the typhoon that occurred in the week and the total precipitation of the week, respectively. This should make it clear what we meant in the next column C = A/B, the contribution of typhoon precipitation to the weekly*
30 *precipitation. We also added notation that "Precipitation was recorded at a 5-min interval at the two rain gauge stations and aggregated to weekly and typhoon precipitation."* Note that this table is now moved to supplementary information as Table S1.

Table S1. The basic information of the typhoon affected weeks.

| Name of typhoons | Date | Precipitation between the first and last typhoon warnings (mm, A) | Total Precipitation of the typhoon week (mm, B) | C = A/B (%) | Discharge between the first and last typhoon warnings (mm, D) | Total discharge of the typhoon week (mm, E) | F = D/E (%) |
|---|---|---|---|---|---|---|---|
| Jelawat | 26-29 Sep. 2012 | 181 | 184 | 98 | 106 | 132 | 80 |
| Soulik | 12-14 Jul. 2013 | 334 | 345 | 97 | 212 | 256 | 83 |
| Trami | 20-22 Aug. 2013 | 296 | 321 | 92 | 230 | 252 | 91 |
| Kong-Rey | 28-30 Aug. 2013 | 258 | 286 | 90 | 187 | 205 | 91 |
| Usagi | 20-22 Sep. 2013 | 265 | 304 | 87 | 159 | 195 | 82 |
| Fitow | 5-7 Oct. 2013 | 305 | 324 | 94 | 191 | 229 | 83 |
| Matmo | 22-23 Jul. 2014 | 170 | 184 | 92 | 145 | 167 | 87 |
| Fung-Wong | 20-23 Sep. 2014 | 180 | 187 | 96 | 112 | 120 | 93 |
| Chan-Hom | 9-11 Jul. 2015 | 243 | 252 | 96 | 157 | 185 | 85 |
| Soudelor | 7-9 Aug. 2015 | 470 | 507 | 95 | 380 | 445 | 85 |
| Goni | 21-24 Aug. 2015 | 160 | 168 | 95 | 112 | 122 | 92 |
| Average | | 260 | 278 | 94 | 181 | 210 | 86 |

1. The accumulated precipitation of typhoons were summed from first and last typhoon warnings issued and the total precipitation (mm) of typhoon week is the average value of two rain gauges (COA530 and COA540) during the week of typhoon influenced, and same as the discharge (the average value of four watersheds, A1, A2, F1, and F2). Precipitation was recorded at a 5-min interval at the two rain gauge stations and aggregated to weekly and typhoon precipitation.

5. P4 L4-18: It would be important to have more basic hydrologic data in this paragraph. For starters, I am missing streamflow hydrographs for the study periods. For a better overview of the general hydrologic conditions. This paragraph and Table 1 contain data to calculate typhoon runoff ratios (the amount of precipitation that becomes runoff over a period of time: streamflow/precipitation), but it would also be interesting to see annual runoff ratios for the watersheds with and without the typhoon periods. Assessing pre-typhoon conditions might also be helpful for interpreting the different response between typhoon and non-typhoon periods but also the variability within the typhoon responses.

Reply: We added hydrograph for the study periods and added the following sentence to describe it. "*During the sampling period, weekly precipitation ranged from 1 mm to 507 mm while weekly streamflow ranged from 10 mm to 446 mm (Fig. 2a and Table S1)*". We also calculated weekly runoff ratio and the mean runoff ratio for the typhoon and non-typhoon periods. In the Results section, we added "*The weekly runoff ratio was negatively related to precipitation quantity and was highly variable during the non-typhoon period but varied much less during the typhoon period (Fig. 2b)*". In the Discussion section, we added:
*"4.1 Differences between typhoon and non-typhoon periods*
*The striking differences in the discharge-budget patterns between typhoon and non-typhoon periods should be related to changes in the relative proportion of sources of stream discharge. Stream discharge originates from three sources, surface runoff, subsurface runoff and groundwater. Among the three sources, groundwater was more important during low than high flow periods, whereas the contribution from surface runoff should be more important during heavy storms than small storms. The contribution from subsurface flow probably dominated the discharge at our study site, especially in F1 and F2 because a study at a natural forest 12 km Southeast from our study site indicated that even during a heavy typhoon storm, with precipitation near 700 mm in two days, there was no observable surface runoff (Lin et al., 2011). The contribution from groundwater and subsurface runoff to total discharge likely resulted in the very high runoff ratios for weeks with small amount of precipitation. For example, in 28 January 2014, the weekly precipitation and discharge were 1.5 mm and 13 mm, respectively, which led to the highest runoff ratio, 8.7, for the entire study period (Fig. 2).*

*Changes in relatively contributions of different sources of water (or old water relative to new water) on stream discharge, play a key role in regulating ion concentrations during a storm and between periods of different flow rates (Elwood and Turner, 1989; Giusti and Neal 1993; Bishop et al., 2004). Among the three sources, groundwater is enriched with ions derived from rock weathering such as $K^+$, $Ca^{2+}$, and $Mg^{2+}$, pre-storm subsurface runoff have a longer contact time with soils that are also rich in these cations and $SO_4^{2-}$, and surface runoff largely reflect precipitation chemistry. A study of storm solute transport in a forested watershed in northern Taiwan, 12 km Southwest of our study site, indicated that*

*concentrations of $Na^+$, $Ca^{2+}$, $Mg^{2+}$, $Cl^-$, and $SO_4^{2-}$ were diluted due to the mixing of large quantities whereas concentrations of $K^+$, $NH_4^+$ and $NO_3^-$ were enhanced during high flows (Wang et al., 1996). In our study, the greater contributions from groundwater and subsurface runoff in the non-typhoon period likely contributed to the greater (more positive) slopes between discharge and budget of many ions for the non-typhoon period than typhoon period, in which many of the relationships were not significant (Figs. 4 and 5). The second possible reason for the greater slopes between discharge and budget of many ions during the non-typhoon period is the differences in ion concentration between typhoon and non-typhoon storms. The day or two days before a typhoon typically has clear sky because the outskirt air masses of the typhoon "blow" away most air pollutants. As a result, precipitation associated with typhoons have low concentrations of ions with terrestrial sources (Lin et al., 2011). In our study, mean concentrations of all ions were lower during typhoon period than non-typhoon period (Table S3) and this diluted precipitation ion concentrations overrode quantity effect and contributed to the smaller increases in budget with increasing discharge in the typhoon period than the non-typhoon period (Figs. 4 and 5)."*

The new Figure 2 is presented in our reply to the general comment.

6.  P4 L17 and Figure 2: Record over what time? What are the "5, 8, 9, and 9 typhoon weeks" and how are they shown in Figure 2? What do the dashed lines in Figure 2 represent?
Reply: We rephrased/corrected the sentence to clarify what we meant to say. The new sentence is now "*Among the 10 highest hourly, 6-hr, 12-hr and 24-hr precipitation events, 5, 8, 9, and 9 of them occurred during weeks associated with typhoon storms.*"

7.  P4 L14-18: This whole paragraph is about hourly intensities, not precipitation totals.
Reply: Yes, indeed this paragraph is about how intense typhoon storms were. The information about precipitation totals is given in the paragraph before this paragraph. Because hydrological processes may be different between intense storms and regular storms, we thought it is important to describe how intense the typhoon storms were.

8.  Table 2: It's good to list the regression models but this could be supplemental information in my opinion
Reply: Table 2 is now moved to supplemental information.
* * *
**Comments from Reviewer #2,**

This manuscript presents relationships between water budget (precipitation and discharge) and weekly hydrochemical data in four watersheds within the Feitsui Reservoir Watershed (FRW) in northern Taiwan. The dataset spans three years and encompasses eleven Typhoon systems. The authors use these data to draw inference about distinct hydrochemical response during typhoon vs. non-typhoon times, both in terms of variability and direction. Additionally, the watersheds differ in size and relative proportion of agricultural land - in this case largely heavily fertilized tea plantations – and they use this difference to examine the effect of land use change on response to typhoon events.

The authors have an interesting and appropriate dataset to address their questions, and additionally seem to have chosen an ideal location to elucidate dependence of hydrochemical response on storm intensity. Their methods are logical and results indicate striking differences between watershed response during typhoon and non-typhoon times. I have two suggestions for review I would characterize as "major," along with multiple minor suggestions and comments, which are delineated below. Given response to these suggestions, this paper seems like a good candidate for publication in Biogeosciences.

1. Discharge estimation and enhanced hydrologic analysis: The author's use of the "area-ratio" method of estimating discharge is far from ideal, although it seems to be unavoidable in the absence of other gauging stations within the study watersheds. It is surprising they did not at least perform weekly in-situ measurements of discharge (area-velocity or dilution gauging) in conjunction with their chemistry sampling. Additionally, they present no hydrography or additional hydrologic analyses, for example runoff ratio at annual and storm scales, which may help interpret their weekly data and results.

1a. I would like to see a more thorough explanation of both their method for estimating discharge (rather than requiring the reader to reference to one of the author's previous publications) and a discussion of the method's limitations. This should not be long or intensive, but should be sufficient to help the reader understand the reasons for doing so and the potential effects on the analysis.

Reply: We agree that the area ratio method is not ideal and re-estimated stream discharge using the Hydrologiska Byråns Vattenbalansavdelning (HBV) model. We updated all the figures and tables based on the new calculations. The basic patterns do not change but some details are different. The following detail is added to the Materials and Methods section.

*"Stream discharge of the four ungauged watersheds was also simulated by the HBV model processed through TUWmodel (ver. 0.1-8) (Parajka et al., 2013). Five daily rain gauges, maintained by Water Resource Agency (WRA), and five metrological stations, maintained by the Central Weather Bureau (CWB) of Taiwan with hourly observed rainfall, temperature,*

*wind speed, and solar radiation were used to estimate daily rainfall and potential evapotranspiration. The daily evapotranspiration is also observed by Taipei Feitsui Reservoir Administration (TFRA, Taiwan) at the Feitsui meteorological station. The observed rainfall, temperature and evapotranspiration were applied into 20 sub-catchments with Thiessen polygon method. Daily discharge was monitored in three main tributaries of Baishi Creek by TFRA. In the calibration against the observed values, parameters were generated by the package DEoptim (ver. 2.2-4) (Mullen et al., 2011). Three objective functions, Nash Efficient Coefficient (NSE), its power of 2 and log scale, were used to adjust the model to suit normal, extreme, and low flow conditions. The validation gauge is located in the inflow of dam of reservoir. The modelled daily discharge was aggregated into weekly discharge.”* We updated all the figures and tables based on the new calculations. The basic patterns do not change but some details are different. The cited references are listed below.

Parajka, J., Viglone, A., Salinas, R. M., Sviapalan, M., and Blöschl, G.: Comparative assessment of predictions in ungauged basins - Part 1: Runoff-hydrograph studies, Hydrol. Earth Sys. Sci., 17, 1783–1795, doi:10.5194/hess-17-1783-2013, 2013.

Mullen, K. M., Ardia, D., Gil, D. L., Windover, D., and Cline, J.: "DEoptim: An R Package for Global Optimization by Differential Evolution, J. Stat. Softw., 40, 1–26, doi: 10.18637/jss.v040.i06, 2011.

1b. The author's should strongly consider including standard hydrographs coupled with their hyetographs for context, and potentially also include annual and a storm-by-storm analysis of runoff coefficients. The latter may require some time, but would be very valuable in interpreting their dataset from the standpoint of variable runoff generation processes due to intensity. Additional context such as intensity-duration curves with differentially colored typhoon and non-typhoon events would also help make points they make primarily in text. Reply: We added standard hydrographs to the results and calculated weekly runoff ratios and examined their relationship to precipitation. The following information is added to the revision. “*During the sampling period, weekly precipitation ranged from 1 mm to 507 mm while weekly streamflow ranged from 10 mm to 446 mm (Fig. 2a and Table S1)*”. We also calculated weekly runoff ratio and the mean runoff ratio for typhoon period and non-typhoon period. In the Results sction, we added “*The weekly runoff ratio was negatively related to precipitation quantity and was highly variable during the non-typhoon period but varied much less during the typhoon period (Fig. 2b)*”.

[Figure]

Figure 2. Mean weekly precipitation, discharge and runoff (a), and the relationship between mean weekly precipitation and mean runoff ratio (b) of the four studied watersheds combined. MAP: Mean annual precipitation, MAS: Mean annual stream discharge.

2. Process-based discussion: With the richness of their chemical dataset, I expected the discussion to be an ideal opportunity to discuss the observed non-linearities in hydrochemical response in the context of physical and biogeochemical process. The authors rightly make no claim at their ability to rigorously distinguish the physical or biogeochemical processes responsible for the dynamics the observe, but there is a rich literature within watershed hydrology and biogeochemistry addressing each of these chemical constituents which would appropriately be used to contextualize their findings. What do the nutrient dynamics suggest about stream response to extreme events? Does stream productivity or nutrient saturation factor in to the differences between A and F watersheds? Does the response of weathering products suggest activation of distinct flowpaths in non-typhoon vs typhoon events? Any of these or more would be appropriate discussion points and would help the reader move past observation to interpretation.
Reply: We substantially extended our discussion to include possible explanations to the differences in the patterns between typhoon and non-typhoon periods. The following is added to the discussion. In the Discussion section, we added:
*"4.1    Differences between typhoon and non-typhoon periods*
*The striking differences in the discharge-budget patterns between typhoon and non-typhoon periods should be related to changes in the relative proportion of sources of stream discharge. Stream discharge originates from three sources, surface runoff, subsurface runoff and groundwater. Among the three sources, groundwater was more important during low than high flow periods, whereas the contribution from surface runoff should be more*

*important during heavy storms than small storms. The contribution from subsurface flow probably dominated the discharge at our study site, especially in F1 and F2 because a study at a natural forest 12 km Southeast from our study site indicated that even during a heavy typhoon storm, with precipitation near 700 mm in two days, there was no observable surface runoff (Lin et al., 2011). The contribution from groundwater and subsurface runoff to total discharge likely resulted in the very high runoff ratios for weeks with small amount of precipitation. For example, in 28 January 2014, the weekly precipitation and discharge were 1.5 mm and 13 mm, respectively, which led to the highest runoff ratio, 8.7, for the entire study period (Fig. 2).*

*Changes in relatively contributions of different sources of water (or old water relative to new water) on stream discharge, play a key role in regulating ion concentrations during a storm and between periods of different flow rates (Elwood and Turner, 1989; Giusti and Neal 1993; Bishop et al., 2004). Among the three sources, groundwater is enriched with ions derived from rock weathering such as $K^+$, $Ca^{2+}$, and $Mg^{2+}$, pre-storm subsurface runoff have a longer contact time with soils that are also rich in these cations and $SO_4^{2-}$, and surface runoff largely reflect precipitation chemistry. A study of storm solute transport in a forested watershed in northern Taiwan, 12 km Southwest of our study site, indicated that concentrations of $Na^+$, $Ca^{2+}$, $Mg^{2+}$, $Cl^-$, and $SO_4^{2-}$ were diluted due to the mixing of large quantities whereas concentrations of $K^+$, $NH_4^+$ and $NO_3^-$ were enhanced during high flows (Wang et al., 1996). In our study, the greater contributions from groundwater and subsurface runoff in the non-typhoon period likely contributed to the greater (more positive) slopes between discharge and budget of many ions for the non-typhoon period than typhoon period, in which many of the relationships were not significant (Figs. 4 and 5). The second possible reason for the greater slopes between discharge and budget of many ions during the non-typhoon period is the differences in ion concentration between typhoon and non-typhoon storms. The day or two days before a typhoon typically has clear sky because the outskirt air masses of the typhoon "blow" away most air pollutants. As a result, precipitation associated with typhoons have low concentrations of ions with terrestrial sources (Lin et al., 2011). In our study, mean concentrations of all ions were lower during typhoon period than non-typhoon period (Table S3) and this diluted precipitation ion concentrations overrode quantity effect and contributed to the smaller increases in budget with increasing discharge in the typhoon period than the non-typhoon period (Figs. 4 and 5)."*

Lin, T. C., Hamburg, S. P., Lin, K. C., Wang, L. J., Chang, C. T., Hsia, Y. J., Vadeboncoeur, M. A., McMullen, C. M. C., and Liu, C. P.: Typhoon disturbance and forest dynamics: lessons from a northwest Pacific subtropical forest, Ecosystems, 14, 127–143, doi: 10.1007/s10021-010-9399-1, 2011.

**Table S3**. The mean (one standard deviation) concentrations of ions (mg/l) in precipitation during non-typhoon and typhoon periods.

| | H | Na$^+$ | K$^+$ | Ca$^{2+}$ | Mg$^{2+}$ | NH$_4^+$ | Cl$^-$ | NO$_3^-$ | SO$_4^{2-}$ | PO$_4^{3-}$ |
|---|---|---|---|---|---|---|---|---|---|---|
| **A1** | | | | | | | | | | |
| Non-typhoon | 0.06 (0.06) | 4.15 (8.35) | 0.64 (1.36) | 0.84 (1.36) | 0.59 (1.04) | 0.85 (1.35) | 9.43 (22.9) | 3.97 (6.27) | 5.18 (7.26) | 0.04 (0.15) |
| Typhoon | 0.02 (0.02) | 3.78 (3.51) | 0.43 (0.45) | 0.48 (0.36) | 0.51 (0.41) | 0.12 (0.14) | 6.05 (5.21) | 0.40 (0.70) | 1.54 (1.57) | 0.01 (0.02) |
| **F2** | | | | | | | | | | |
| Non-typhoon | 0.04 (0.06) | 3.37 (8.17) | 0.61 (1.66) | 0.84 (1.91) | 0.55 (1.08) | 0.55 (1.18) | 7.96 (23.9) | 2.56 (9.10) | 4.84 (10.2) | 0.01 (0.05) |
| Typhoon | 0.01 (0.02) | 3.36 (2.75) | 0.37 (0.39) | 0.32 (0.23) | 0.46 (0.36) | 0.14 (0.17) | 5.03 (3.96) | 0.39 (0.53) | 2.11 (2.80) | 0.002 (0.003) |

[Figure]

Figure 4: Relationship between stream discharge and nutrient budget (stream output – precipitation input) of cations ($Na^+$, $K^+$, $Ca^{2+}$, $Mg^{2+}$, and $NH_4^+$). The gray, black, and dash lines indicate significant linear regressions between discharge and ions budgets for non-typhoon, typhoon and all data, respectively. Please refer to Table S2 for the regression models and $R^2$s.

[Figure]

Figure 5: Relationship between stream discharge and nutrient budget (stream output – precipitation input) of anions (Cl$^-$, NO$_3^-$, SO$_4^{2-}$, and PO$_4^{3-}$). The gray, black, and dash lines indicate significant linear regressions between discharge and ions budgets for non-typhoon, typhoon and all data, respectively. Please refer to Table S2 for the regression models and R$^2$s.

See below for minor suggestions and comments delineated by page and line number:

P2 L25-37: More descriptive climatic information would help the portion of the audience not familiar with the location: annual precipitation, climate zone, seasonality, etc. L30-31 could include actual quantification.

Reply: Following the general description, we added the following information: *"The FRW region is characterized with humid subtropical climate. The mean annual precipitation is 3765 mm between 1991 and 2001 (Chen et al., 2006), with approximately 68% occurring between May and September (Chang and Wen, 1997). However, due to the rough topography, precipitation is highly variable, ranging from 3500 mm in the southwest portion of the FRW to more than 5000 mm in the northeast during 2001–2010 (Huang, C. J., unpublished data)."*

Chen, Y. J., Wu, S. C., Lee, B. S., and Hung, C. C: Behavior of storm-induced suspension interflow in subtropical Feitui Reservoir, Taiwan, Limnol. Oceanogr., 51, 1125–1133, doi: 10.4319/lo.2006.51.2.1125, 2006.

Chang, S. P., and Wen, C. G.: Changes in water quality in the newly impounded subtropical Feitsui Reservoir, Taiwan. J. Am. Water Resour. Assoc., 3, 343–357, doi: 10.1111/j.1752-1688.1997.tb03514.x, 1997.

P3 L3-6: This watershed description could go more appropriately in the previous section as study area description.

Reply: We moved this paragraph to 2.1 Study region.

P3 L7: At what frequency? Or was it recorded continuously?

Reply: It was on a weekly basis. We changed the sentence to "*Weekly samples were collected with a 20-cm diameter polyethylene (PE) bucket. Weekly stream water samples were collected by immersing a PE bucket into the stream."*

P3 L9-10: Could the authors explain more about how they handled their samples, particularly with regard to the nutrients? As described their methods may be problematic with regard to nutrients. Particularly in what I would assume is a relatively warm, high productivity system, samples should be field-filtered or chemically stabilized and then frozen as soon as possible. Alternatively they may be analyzed immediately upon return from the field. If they did neither, it might be appropriate to include some discussion of the effect of uptake on their samples. For example, in the lower $NO_3$, F watersheds there may have been more relative uptake after sampling than in the presumably eutrophic, potentially saturated, A watersheds. $NH_4$ is also challenging because it is so readily nitrified.

Reply: The samples were filtered the same day of collection after pH and conductivity measurements. The samples were analyzed within two weeks. Due to the high agricultural export of $NH_4^+$, it is possible that some $NH_4^+$ may be nitrified. However, the much higher concentrations of $NH_4^+$ in A1 and A2 than F1 and F2 (Lin et al., 2015) suggest that nitrification between sample collection and analysis did not lead to changes in the patterns we reported in this study. In addition, a study of nitrogen processes at a mountain watershed 12 km Southeast from our study site indicated that $NO_3^-$ and $NH_4^+$ concentrations were not different between split samples, half left in the site for one week and half brought back to the laboratory and analyzed immediately (i.e., the same day of collection) (Lu et al., 2017). Thus, we believe that the changes in water chemistry between collection and analysis should not be substantial. We added the following to the revision. *"After the measurement of pH and conductivity, samples were filtered (0.45 $\mu$m filter paper) mostly within eight hours of sample collection."*

Lu, M. C., Chang, C. T., Lin, T. C., Wang, L. J., Wang, C. P., Hsu, T. C., Huang, J. C.: Modeling the terrestrial N processes in a small mountain catchment through INCA-N: A case study in Taiwan. Sci. Total Environ., 593, 319-329, doi: 10.1016/j.scitotenv.2017.03.178, 2017.

P3 L15-20: More is needed here. As mentioned above, I'd like to see a better description and justification for the discharge scaling method they used.
Reply: Please see our response to the general comment 1a for a comprehensive response to this comment.

P3 Section 2.4: Here the authors spend significant time discussing another potential source of error; something similar is needed for discharge.
Reply: We changed the method for the estimation of discharge from area ratio to HBV model (please see reply to comment 1a). We also added the following to acknowledge the potential error associated with the estimation. *"Although the HBV model has been successfully applied in northern Taiwan (Chang et al., 2017a), due to the lack of in-situ measurements of discharge, the estimates are subject to some uncertainty."*

Chang, C. T., Wang, L. J., Huang, J. C., Liu, C. P., Wang, C. P., Lin, N. H., Lixin, W., and Lin, T. C.: Precipitation controls on nutrient budgets in subtropical and tropical forests and the implications under changing climate. Adv. Water Resour., 103, 44–50. doi.: 10.1016/j.advwatres.2017.02.013, 2017a.

P4 L20-21: This seems like an interesting finding, rather than differences in regression direction, since many of the regressions for typhoon periods have very low predictability.
Reply: We agree that this is an important finding and that is why we put it at the very beginning of our results following the "Basic storm information".

P4-5 Section 3.3: Reiterating above, the change in direction of the regression is less compelling to me than the dramatic change in spread of the point cloud between non-typhoon and typhoon periods (i.e., much lower predictability).
Reply: Yes, we agree that this is an important finding and that is why we put the unpredictability during typhoon period at the very beginning of our results following the "Basic storm information".

P6 L9-10: I'd be interested in a comparison of extreme, non-typhoon events with comparable typhoon events. Do the authors feel that it is merely the high intensity of the typhoons that cause the unique hydrochemical response? Or is there something unique about typhoons? High winds or variable winds that can damage forests and farms that change delivery of ions to streams?
Reply: We added nutrient budget of large non-typhoon storms into the comparison of nutrient budget (New Fig. 6). We selected weeks with precipitation greater than the minimum typhoon-week precipitation (160 mm). *"The budget of most ions of the seven large non-typhoon storms, with precipitation greater than the minimum typhoon precipitation (160 mm) was between the budget of typhoon weeks and regular non-typhoon weeks, but there were fundamental differences (Fig. 6). For example, the negative budget of $Na^+$, $Cl^-$ and $PO_4^{3-}$ was only observed during typhoon weeks (Fig. 6)."* We also added the following to the discussion. *"The lack of predictability of stream discharge on the budget of several ions is possibly due to damages to the forests and farms by the typhoons. Damages to trees may affect the level of foliar nutrient leaching and nutrient uptake by roots and thus the nutrient export (Lin et al., 2011). The poor correlation between maximum wind velocity and precipitation quantity reported by Lin et al. (2011) suggests that precipitation quantity is not a good predictor of the magnitude of typhoon influences on nutrient input-output budget and likely contributed to the low predictability of discharge on ion budget during typhoon period."*

Lin, T. C., Hamburg, S. P., Lin, K. C., Wang, L. J., Chang, C. T., Hsia, Y. J., Vadeboncoeur, M. A., McMullen, C. M. C., and Liu, C. P.: Typhoon disturbance and forest dynamics: lessons from a northwest Pacific subtropical forest, Ecosystems, 14, 127–143, doi: 10.1007/s10021-010-9399-1, 2011.

P7 L8-10: To reiterate a point from the methods, I would encourage the authors to think through the potential effect of nutrient uptake after sampling, if indeed there was no filtering or other stabilization and there was a period of multiple days before analyzing the samples. If the F and A watersheds have distinct nutrient regimes, it is also reasonable to expect they may exhibit distinct uptake responses, which could differentially affect each set of samples. Reply: We did filter the samples the same day of collection and mostly within 8 hours of collection. Although this could not rule out the possibility of some nutrient uptake by microbes, we do not think this would be substantial. In addition, a study of nitrogen processes at a mountain watershed 12 km Southeast from our study site indicated that $NO_3^-$ and $NH_4^+$ concentrations were not different between split samples, half left in the site for one week and half brought back to the laboratory and analyzed immediately (i.e., the same day of collection) (Lu et al., 2017). Thus, we believe that the changes in water chemistry between collection and analysis should not be substantial. We added the following to the revision. *"After the measurement of pH and conductivity, samples were filtered (0.45 µm filter paper) mostly within eight hours of sample collection."*

Lu, M. C., Chang, C. T., Lin, T. C., Wang, L. J., Wang, C. P., Hsu, T. C., Huang, J. C.: Modeling the terrestrial N processes in a small mountain catchment through INCA-N: A case study in Taiwan. Sci. Total Environ., 593, 319-329, doi: 10.1016/j.scitotenv.2017.03.178, 2017.

P7-8 Section 4.3: The authors provide some interesting, process-based context for their findings in this section! This is the type of discussion I would like to see throughout with respect to all of their findings.
Reply: We have substantially expanded our discussion to include process-based context for the findings, with special focus on the differences between typhoon and non-typhoon periods. Please see our response to general comment #2.

Tables 1 and 2: These could go in a supplementary section
Reply: As suggested, we have moved the two tables to supplementary information.

Figures 2 and 3: The authors did a good job of making their figures interpretable in grayscale by choosing their colors and using open vs closed circles. They could further improve this by doing something similar for the regression lines shown here, perhaps dotted lines for the black, total dataset regressions? Purple and blue lines should be distinguishable in grayscale
Reply: As suggested, we have change the figures to grayscale.
* * *

[revised manuscript text omitted]

---

## Author Response (AR2)

Reviewer #2 Reconsider after major revision

**Suggestions for revision or reasons for rejection (will be published if the paper is accepted for final publication)**

I appreciate the authors' efforts to incorporate the suggestions. A few more comments.

1. Modeling discharge is a viable approach in this case. That being said, it is not clear to me how the model was calibrated. Was the model calibrated on the discharge from the two nearby gauged stations mentioned in the first version of the manuscript? If so, when transferring the calibration parameters to the four ungauged watersheds, were the differences in land use taken into account or was the model run with the same parameter set for all four watersheds? If the same parameter set was used for all four watersheds, then the authors might not gain much from modeling streamflow using HBV vs simply applying the drainage-area ratio method. The authors need to elaborate on the model calibration.

**Reply**:

In the previous revision, as recognized by the reviewer, we used the HBV model and two nearby gauged stations to obtain calibrating parameters and then transferred the calibrated parameters to simulate discharge of the ungauged catchments. Due to the lack of important information such soil infiltration capacity and evaporation of each land use type, it is not possible to include land use differences into the model. Estimating ungauged streamflow is still a daunting subject in hydrology especially in areas that are not easily accessible. Numerous studies have demonstrated the various methods in estimating ungauged streamflow, but we are not aware of any method that is widely considered to be superior than others. In addition, there is no widely accepted method for evaluation of spatial homogeneity, which is a fundamental assumption for streamflow estimation for ungauged catchments. Moreover, without direct measurements of streamflow, it is difficult to confidently validate the estimation even if we could include land use differences into the model. At the same time, the main focus of our study is the shifts in ion flux between non-typhoon and typhoon periods not streamflow estimation (or separation). Moreover, stream flow estimated from the area-ratio method and the HBV model showed similar results and the choice of methods should not affect our main findings of the striking shifts in the relationship between precipitation and ion export between typhoon and non-typhoon periods. Thus, while recognizing that streamflow estimation is an important issue, we believe that accuracy of streamflow during the two flow regimes should be acceptable because the differences between the two periods are distinct.

2. Runoff ratios are included now but it appears they might not be as helpful as I initially thought. First off, the runoff ratios aren't discussed much. The authors state that "The weekly runoff ratio was negatively related to precipitation quantity", which of course does not make much sense. The reason this appears to be the case is-and the authors identify this correctly-because the time interval is very

short and sometimes little precipitation fell in a one-week period, so that runoff from baseflow were larger than the fallen precipitation. On such short time steps the baseflow would have to be removed in order to make the runoff ratio values meaningful. At this stage, the runoff ratio analysis unfortunately does not add much to the paper. I would suspect that with baseflow removed, a clearer separation between typhoon and non-typhoon periods would emerge.

**Reply**:

As suggested by the reviewer, we used the HBV simulated total streamflow and rapid surface runoff to calculate the runoff ratios (Table 1). The result clearly indicates that total runoff ratio was not different between typhoon and non-typhoon periods, as we reported in the previous revision. But after both groundwater and subsurface flow being removed, the (rapid) runoff ratio was greater during the typhoon period (0.27–0.33) than during the non-typhoon period (0.06–0.15), as the Reviewer expected.

Further, to illustrate the effect of differences in runoff patterns on ion export, we also calculated nitrate export via runoff for non-typhoon (n = 123) and typhoon (n = 11) periods. Nitrate was chosen because the relationship between precipitation and nitrate export was strikingly different between forested watersheds (F1 and F2) and watersheds with agricultural lands (A1 and A2). For both typhoon and non-typhoon periods, the riverine $NO_3^-$ concentrations in A1 and A2 were always considerably higher than that in precipitation (Figure 1). The watersheds with agricultural lands exported more nitrate (Figure 1) likely due to the excessive N input from fertilizers. The riverine nitrate concentrations in A1 and A2 were similar between non-typhoon and typhoon periods (Figure 1). Because surface runoff was more dominant during typhoons, the similar riverine nitrate concentrations between typhoon and non-typhoon periods may indicate that the soil profile stores large quantity of nitrate available for storm flushing. By contrast, the nitrate concentrations in F1 and F2 were lower than that in precipitation during non-typhoon period (Figure 1) indicating the watersheds retained some nitrates input from rainfall. During the typhoon period, the riverine nitrate concentration was slightly higher than that in precipitation (Figure 1). The contrasting differences in nitrate concentration between non-typhoon period and typhoon period for F1 and F2 but not A1 and A2 highlights the effects of fertilization on watershed nutrient export.

We revised the corresponding paragraphs in the second revision in accordance with the above reply. However, as noted in our reply to the first comment, as constrained by the data availability (e.g., direct measurements of streamflow), we cannot make strong arguments on these mechanisms but missing these details won't affect our overall conclusions regarding the shifts in ion flux between non-typhoon and typhoon periods. Thus, the differences in nitrate concentration between the two periods of the four watersheds described above were not included in the revised manuscript (Fig. 3).

Specifically, in reply to the second comment, we added the following to the revision with blue text. Methods:

*The total runoff derived from the HBV model was further separated into three components, surface runoff, subsurface runoff and groundwater.*

Results:

*The ratio of total runoff to precipitation was not different between non-typhoon period (0.69–0.81) and the typhoon period (0.64–0.78) but the ratio of surface runoff to precipitation was smaller in the non-typhoon period (0.06–0.15) than the typhoon period (0.27–0.33) (Table 1) because proportionally surface runoff was greater during the typhoon period than the non-typhoon period (Fig. 3).*

Discussion:

*The effect of subsurface runoff and groundwater on disrupting the precipitation-runoff relationship is evident from the greater ratio of surface runoff to precipitation during the typhoon period than the non-typhoon period while the ratio of total runoff to precipitation was not different between the two periods (Table 1 and Fig. 3). Our results indicate that under certain circumstances, contributions from baseflow need to be removed in order to detect and meaningfully assess the precipitation-runoff relationship (Table 1). However, it is noted that without direct measurements of streamflow in two of the watersheds, it is difficult to confidently validate the estimation of streamflow separation in this case.*

**Table 1.** Runoff ratio for the four watersheds during typhoon and non-typhoon periods.

| Watersheds | Typhoon period | | Non-typhoon period | |
| --- | --- | --- | --- | --- |
| | Surface runoff : Precipitation | Total runoff : Precipitation | Surface runoff : Precipitation | Total runoff : Precipitation |
| A1 | 0.29 | 0.69 | 0.12 | 0.80 |
| A2 | 0.33 | 0.64 | 0.15 | 0.69 |
| F1 | 0.27 | 0.69 | 0.14 | 0.76 |
| F2 | 0.33 | 0.78 | 0.06 | 0.81 |

[Figure]

Figure 1. The average weekly precipitation and estimated streamflow composition during typhoon and non-typhoon periods among the four watersheds. The red triangles and squares indicated the average weekly $NO_3^-$ concentration of precipitation and streamwater, respectively. The error bars indicate one standard deviation.

[revised manuscript text omitted]